# Multifaceted regulation of siderophore synthesis by multiple regulatory systems in *Shewanella oneidensis*
Peilu Xie[1,2], Yuanyou Xu[1,2], Jiaxin Tang[1], Shihua Wu [1] ✉ & Haichun Gao [1] ✉

Siderophore-dependent iron uptake is a mechanism by which microorganisms scavenge and utilize iron for their survival, growth, and many specialized activities, such as pathogenicity. The siderophore biosynthetic system PubABC in *Shewanella* can synthesize a series of distinct siderophores, yet how it is regulated in response to iron availability remains largely unexplored. Here, by whole genome screening we identify TCS components histidine kinase (HK) BarA and response regulator (RR) SsoR as positive regulators of siderophore biosynthesis. While BarA partners with UvrY to mediate expression of *pubABC* post-transcriptionally via the Csr regulatory cascade, SsoR is an atypical orphan RR of the OmpR/PhoB subfamily that activates transcription in a phosphorylation-independent manner. By combining structural analysis and molecular dynamics simulations, we observe conformational changes in OmpR/PhoB-like RRs that illustrate the impact of phosphorylation on dynamic properties, and that SsoR is locked in the 'phosphorylated' state found in phosphorylation-dependent counterparts of the same subfamily. Furthermore, we show that iron homeostasis global regulator Fur, in addition to mediating transcription of its own regulon, acts as the sensor of iron starvation to increase SsoR production when needed. Overall, this study delineates an intricate, multi-tiered transcriptional and post-transcriptional regulatory network that governs siderophore biosynthesis.

Iron is essential for virtually all organisms due to its involvement in a range of fundamental biochemical processes such as electron transfer, metabolism, amino acid and nucleoside synthesis, DNA synthesis, photosynthesis, and gene expression[1–3]. Despite the abundance of iron in the environment, iron acquisition remains a formidable challenge to microorganisms since free iron is readily oxidized to the ferric state and can form insoluble ferric hydroxide polymers under aerobic conditions[4]. To overcome this, microbes have evolved sophisticated mechanisms to obtain iron in various forms from their surroundings, including ferrous ($Fe^{2+}$), ferric ($Fe^{3+}$), and iron-containing organic molecules, such as heme. Consistently, transport systems dedicated to iron uptake are many and diverse[3,5]. Among them, siderophore-dependent iron acquisition systems are particularly effective in scavenging iron from environmental stocks[1,6].

Siderophores are a chemically diverse group of secondary metabolites that bind iron with high affinity, forming soluble $Fe^{3+}$-siderophore complexes that can be subsequently taken up into the cell[7,8]. Given the critical role of siderophores in iron uptake, their biosynthesis and transport are subject to tight regulation. In many bacterial species, the ferric uptake regulator (Fur) is a key player in sensing intracellular iron levels and modulating gene expression related to siderophore biology[1]. Additionally, some two-component systems (TCSs) have been implicated in governing the synthesis and transport of siderophores, such as AlgZ/AlgR and GacS/GacA of *Pseudomonas aeruginosa*[9,10]. A prototypical TCS contains histidine kinase (HK), which typically is membrane-bound, and soluble cytoplasmic response regulator (RR). The HK, upon detecting an environmental stimulus, undergoes auto-phosphorylation and subsequently transfers the phosphoryl group to its cognate RR[11]. The phosphorylation in the RR at a conserved aspartate residue induces a conformational change, altering the activity of its effecting domain. RRs are most often DNA-binding proteins that function as a transcriptional regulator[12]. Apart from TCSs, regulators of other types that play a non-negligible role in the regulation of siderophore pathway have been known in diverse bacteria, such as sigma factor (e.g., PvdS), sRNA (e.g., RhyB) and RNA chaperone (e.g., Hfq)[13–15].

---

[1]Institute of Microbiology and College of Life Sciences, Zhejiang University, Hangzhou, Zhejiang 310058, China. [2]These authors contributed equally: Peilu Xie, Yuanyou Xu. ✉e-mail: drwushihua@zju.edu.cn; haichung@zju.edu.cn

Many environmental bacteria are renowned for their respiration versatility, which is in large part due to a vast number of iron-containing proteins, especially hemoproteins[16,17]. One of the best-studied examples is *Shewanella*, a group of γ-proteobacteria capable of utilizing numerous compounds as terminal electron acceptors, including oxygen, fumarate, diverse organic and inorganic nitrogen and sulfur compounds, iron, and other metals[18–20]. Conceivably, these bacteria require iron in substantially larger quantities than model organisms, such as *Escherichia coli*[18]. Most *Shewanella*, as the genus representative *S. oneidensis*, possess a three-gene operon (*pubABC*) for the only enzymatic system catalyzing synthesis of three natural macrocyclic hydroxamate siderophores, with putrebactin as the predominant species and avaroferrin as a robust inhibitor of bacterial swarming behavior[21–23]. Moreover, this PubABC system is rather relaxed in substrate specificity, capable of producing numerous siderophores if proper synthetic precursors are available[24].

Despite the importance of the PubABC system in physiology and ecology of *S. oneidensis* as well as its great potential in biotechnology and pharmaceutical industry, how the system is regulated at transcriptional levels and beyond remains largely unknown although Fur and SO_2426 have been implicated[8,25,26]. Here, by using transposon mutagenesis, we identified TCS components BarA and SO_2426 (renamed as SsoR for siderophore synthesis orphan regulator) as crucial regulatory systems that impact siderophore production. By partnering with UvrY, BarA mediates siderophore synthesis through two small RNAs (*CrsB1* and *CrsB2*) and the RNA-binding protein CsrA via posttranscriptional regulation. In contrast, SsoR functions as an orphan RR, and strikingly its regulatory activity was found to be independent of phosphorylation. Structural analysis and molecular simulations reveal that SsoR exists in one form only, which mimics the phosphorylated state observed in phosphorylation-dependent RRs. Furthermore, we showed that Fur senses iron levels and regulates transcription of the *pub*

operon and *ssoR*. In summary, by illustrating a complex and multilayered regulatory network of siderophore synthesis, our results shed light on the evolution of siderophore production system and its regulation in bacteria.

## Results

### BarA and SsoR are positive regulators of siderophore synthesis

This study aimed to identify potential regulators involved in regulation of siderophore synthesis in *Shewanella*. We took advantage of an unexpected color-loss phenotype of Δ*putA* when grown on LB agar plates, which lacks siderophore receptor PutA[27]. Typically, colonies and cell pellets of the *S. oneidensis* wild-type (WT) strain are reddish-brown, a consequence of the unusually abundant cytochrome *c* (cyt *c*) proteins (Fig. 1a). On the contrary, the Δ*putA* strain loses this signature color, indicative of a significantly reduced cyt *c* content (Fig. 1a). This phenotypic change is attributed to iron shortage, a result of the enhanced production, secretion, and extracellular accumulation of siderophore (Fig. 1a)[28]. Conceivably, the phenotype can be suppressed by the removal of siderophore synthesis operon *pubABC*. The Δ*putA*Δ*pub* strain, the same as Δ*pub* (deleting all three *pub* genes), has a WT-level cyt *c* content and consequently regains reddish-brown color (Fig. 1a). To screen for genes affecting siderophore biosynthesis, a transposon library was constructed from the Δ*putA* strain and many colonies that recovered reddish-brown color were obtained (Supplementary Fig. 1a). While most of the suppressor mutants carried transposon sequence in the *pub* operon as expected, several had insertions mapped to *barA* and *ssoR* genes multiple times, which encode the HK of BarA/UvrY TCS and an orphan RR respectively (Fig. 1b). Siderophore assays verified that siderophore production in these isolates was significantly compromised, and this observation was further substantiated with *barA* and *ssoR* in-frame deletion mutants as well as genetic complementation (Fig. 1c, Supplementary Figs. 1b, 2).

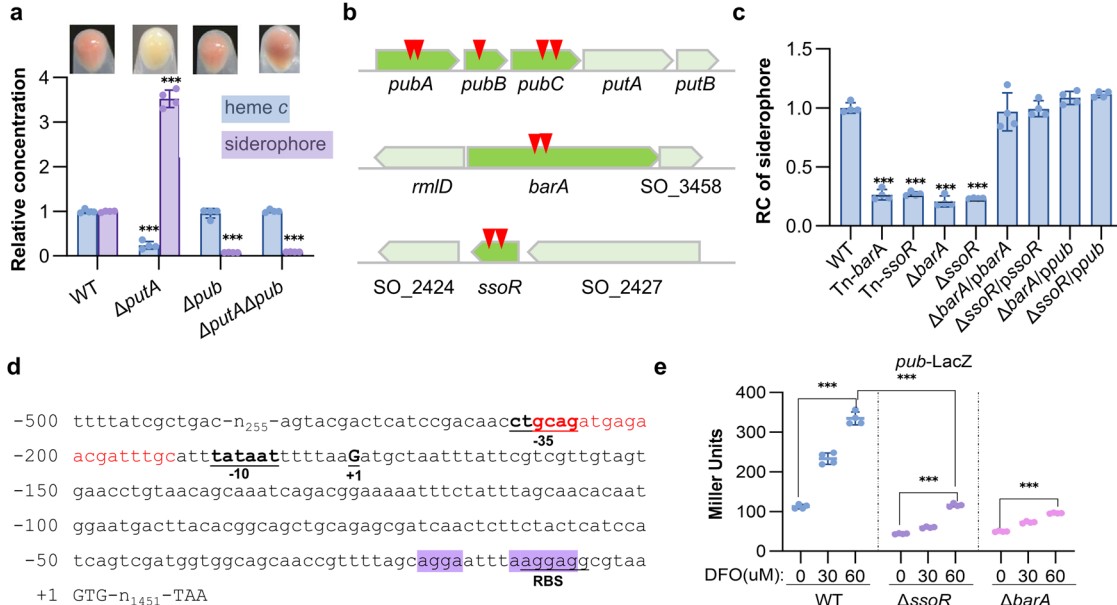

**Fig. 1 | BarA and SsoR are positive regulators of siderophore synthesis. a** Colors of cell pellets and contents of heme *c* and siderophore. Mid-exponential phase cultures ($OD_{600}$, ~0.6) of *S. oneidensis* strains grown in LB were pelleted for photograph and then used for heme *c* content measurement, with the supernatants used for siderophore concentration measurement. The data were first adjusted by the protein levels of the samples, and then the averaged levels of heme *c* and siderophore of the mutants were normalized to that in the wild-type (WT) strain, which was set to 1, giving relative concentration (RC). **b** Genomic context of the *pubABC*, *barA*, and *ssoR* loci in *S. oneidensis* with the transposon insertion sites marked with red triangles. **c** Siderophore production of the *barA* and *ssoR* defective and complementary strains. Complementation was carried out with a vector containing IPTG-inducible

promoter $P_{tac}$. Results shown were from 0.2 mM IPTG. **d** The leader region of the *pub* operon (−500 to +3 relative to the translation start codon). The predicted −10 and −35 box and ribosome binding site (RBS) are underlined. G(+1) represents the transcription start site. The GGA motifs of two potential CsrA-binding sites are highlighted in purple. The Fur-binding motif is in red. **e** Expression of the *pub* operon revealed by *pub*-LacZ fusion in relevant strains grown under iron-repleted or -depleted conditions. Cells of the mid-exponential phase were collected for β-galactosidase activity assay. Data were presented either as means ± SEM. Student's *t* test was performed for statistical analysis between indicated strain and WT under indicated conditions, or between marked samples, $n = 4$ biologically independent experiments, *$p < 0.05$, **$p < 0.01$, ***$p < 0.001$.

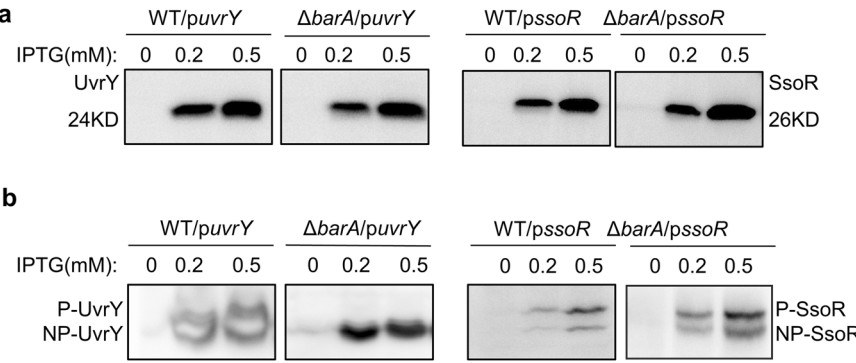

**Fig. 2 | BarA partners with UvrY but not with SsoR. a** Conventional SDS-PAGE and Western blotting analysis of UvrY and SsoR. His$_6$-tagged UvrY and SsoR proteins were inducibly expressed in WT and $\Delta barA$ strains. Proteins were extracted from mid-exponential phase cells, separated on 10% SDS-PAGE, Transferred to PVDF, probed with mouse monoclonal His$_6$-tag antibody, and detected by chemiluminescence. **b** Mn(II)-Phos-tag SDS-PAGE and western blotting of UvrY and SsoR. The same protein preparations were separated on 10% SDS-PAGE containing 50 µM acrylamide-pendant Phos-tag ligand and 100 µM MnCl$_2$. The phosphorylated UvrY (P-UvrY) and SsoR (P-SsoR) proteins moved slower on the gel due to the attached Phos-tag than non-phosphorylated counterparts (NP-UvrY and NP-SsoR).

Influence of BarA and SsoR in vivo on the expression of the *pub* operon was then accessed by monitoring the activity of a chromosomally integrated LacZ reporter (*pub*-LacZ) fused to the leader region of *pub* operon (−500 to the translation start codon) in cells grown under iron-repleted and iron-depleted conditions (Fig. 1d). While LB broth is used as an iron-repleted medium, desferrioxamine (DFO), which is a commercially available siderophore that cannot be imported into *S. oneidensis* cells[28], was supplemented to LB to create an iron-depleted medium. As expected, *pub* expression in the WT strain was substantially induced, by more than 3-fold, under iron-depleted conditions, and this induction was also observed in the absence of either BarA or SsoR (Fig. 1e). Importantly, the *pub* expression in the $\Delta barA$ and $\Delta ssoR$ strains were significantly lower than that in the WT strain under all conditions tested (Fig. 1e). Moreover, the reduced siderophore production in the $\Delta barA$ and $\Delta ssoR$ strains was found to be restored to WT levels by enforced *pub* expression to proper levels (Fig. 1c, Supplementary Fig. 2). Collectively, these data conclude that BarA and SsoR act as positive regulators for siderophore synthesis.

## BarA partners with UvrY but not with SsoR

BarA is the HK of a highly conserved TCS called BarA/UvrY (also referred to as GacS/GacA, BarA/SirA, etc in various species), which has been intensively studied in many Gram-negative bacteria[29–32]. BarA is a tripartite HK that has two N-terminal transmembrane domains followed by a cytoplasmic HAMP domain, a histidine kinase A domain, an ATPase domain, a receiver domain, and a C-terminal histidine phosphotransfer (Hpt) domain (Supplementary Fig. 3)[30]. Both SsoR and UvrY comprise an N-terminal CheY-like receiver (REC) domain but differ from each other in the C-terminal DNA-binding domain: an OmpR/PhoB type winged helix-turn-helix (wHTH) in the former versus a LuxR/FixJ type helix-turn-helix (HTH) in the latter (Supplementary Fig. 3)[33,34]. The phosphorylation residue of *S. oneidensis* UvrY is Asp54 (D54)[30], and its counterpart within SsoR is highly likely to be Asp52 (D52) according to the annotation of the Uniprot database. BarA in *S. oneidensis*, the same as in all other bacteria hosting the TCS, is regarded as an orphan HK because it is not in proximity with *uvrY* on the chromosome. Not surprisingly, cross-talk between BarA and non-cognate RRs (CusR, NarL, NarP, YgeK, RcsB) has been reported[35–37]. Given that both BarA and SsoR but no other TCS components were identified by transposon screening, we speculated that BarA may function as the HK for SsoR too.

To test this, we expressed C-terminally His$_6$-tagged SsoR and UvrY under the control of isopropyl β-D-1-thiogalactoside (IPTG)-inducible promoter P$_{tac}$ in WT and $\Delta barA$. Total proteomes were extracted from cells grown to the mid-exponential phase and applied to SDS-PAGE containing phosphate-binding tag (Phos-tag), which can associate with the divalent cation of Mn$^{2+}$ and form a complex with the phosphorylated proteins, thus

retarding migration[38]. Conventional SDS-PAGE followed by Western blotting revealed single bands for both UvrY and SsoR recombinant proteins regardless of the strain background, with band intensities correlating with IPTG concentrations (Fig. 2a). However, these two proteins behaved clearly differently in Phos-tag SDS-PAGE and Western Blotting (Fig. 2b). UvrY proteins existed in both the phosphorylated and unphosphorylated forms in WT, but only in the unphosphorylated form in the absence of BarA. In contrast, SsoR proteins were always present in both forms. These data indicate that phosphorylation of UvrY but not SsoR in vivo is dependent on BarA, and therefore, BarA/UvrY and SsoR regulate siderophore synthesis through separate pathways in *S. oneidensis*.

## BarA/UvrY TCS positively regulates siderophore synthesis through the Csr regulatory cascade

The signals sensed by BarA have been suggested to be metabolic end products, short-chain fatty acids in particular, such as formate and acetate[39]. Subsequently, a classical phosphor-relay occurs, resulting in phosphorylated UvrY (UvrY-P), which in turn activates the transcription of small regulatory RNAs CsrB and CsrC[40]. These RNAs interact directly with CsrA, a global RNA-binding protein, influencing its ability to either repress or enhance the expression of its RNA targets[41,42], thereby affecting diverse biological processes, including carbon metabolism, biofilm formation, motility, virulence, and siderophore synthesis[31,43–47].

The BarA regulatory cascade of *S. oneidensis* has been proposed to include BarA, UvrY, two regulatory RNAs CsrB1 and CsrB2, and presumably CsrA[30]. To unravel how BarA/UvrY/Csr system is linked to siderophore synthesis in *S. oneidensis*, we assessed siderophore levels in mutants lacking each of these components. It was worth mentioning that the $\Delta csrA$ strain showed extremely severe growth defects when grown in LB, which could be completely corrected by moderate expression of *csrA* in trans (Supplementary Fig. 4a), echoing that the loss of CsrA has profound and pleiotropic effects on the physiology of *E. coli*[48]. Like $\Delta barA$, the $\Delta uvrY$ strain was heavily defective in siderophore production (Fig. 3a). Similar results were obtained from a *csrB1csrB2* double knockout (Fig. 3a), which was expected as the expression of *csrB1* and *csrB2* depends on BarA/UvrY (Fig. 3b). Expression of either *csrB1* or *csrB2* in trans in $\Delta csrB1\Delta csrB2$ was able to restore siderophore synthesis (Supplementary Fig. 2), indicating that both sRNAs are functional. Conceivably, similar to the effect of enforced expression of the *pub* operon, enforced expression of either *csrB1* or *csrB2* in the $\Delta uvrY$ strain led to a substantial increase in siderophore production (Fig. 3c, Supplementary Fig. 2). We also observed that siderophore production was inversely correlated to CsrA levels. The *csrA* deletion drastically increased siderophore production, and this effect was independent of BarA/UvrY (Fig. 3a, Supplementary Fig. 1b), whereas overexpression of *csrA* in the

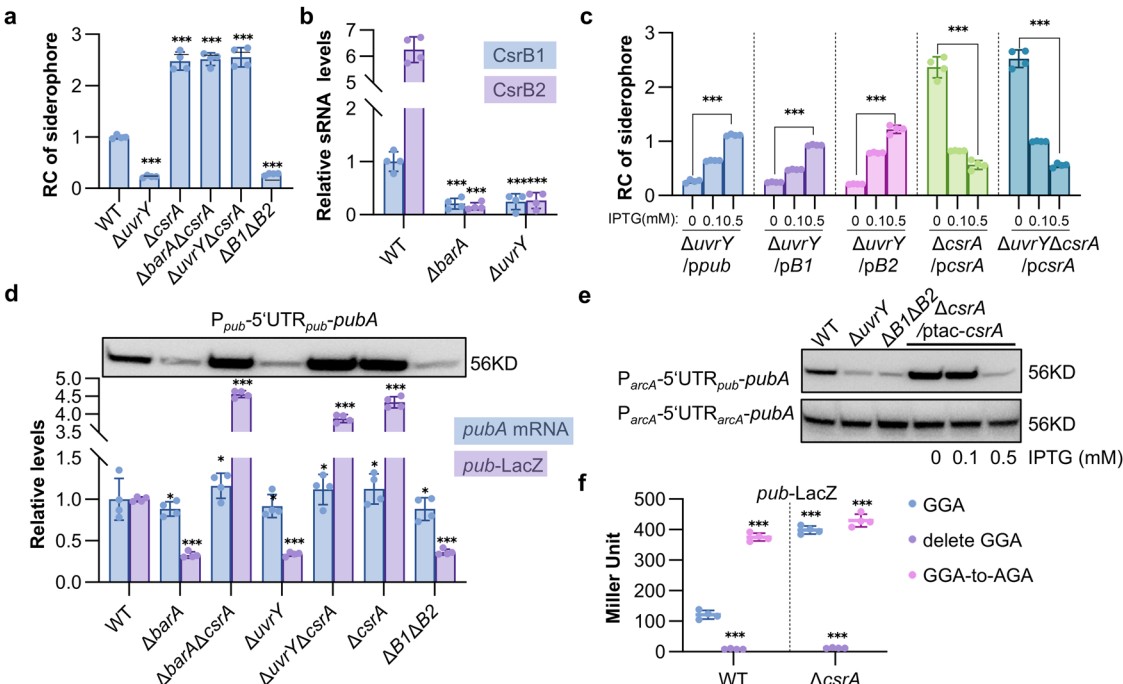

**Fig. 3 | BarA/UvrY TCS positively regulates siderophore synthesis through the Csr regulatory cascade. a** Siderophore production of indicated *S. oneidensis* strains. **b** The RNA levels of *csrB1* and *csrB2* by qRT-PCR. The averaged values for each transcript were normalized to that of the *csrB1* in WT. **c** Effects of CsrB1, CsrB2, and CsrA on the production of siderophore. **d** Transcription and translation levels of *pub*. The ratios of transcription levels of *pubA* revealed by qRT-PCR and translation levels revealed by *pub*-LacZ reporter between WT and the indicated mutant strains were shown. Upper panel, PubA (His$_6$-tagged) levels in indicated strains determined by Western blotting. The protein was expressed within pHG101 driven by the

natural leader region (5'UTR) of the *pub* operon. **e** PubA (His$_6$-tagged) levels driven by different leader regions. The protein was expressed within pHG101 driven by the *arcA* promoter combined with 5'UTR of *pub* or *arcA*. **f** Activity of the *pub*-LacZ fusion with deletion GGA mutation and GGA-to-AGA point mutation of CsrA-binding site (AAGGAG) in WT and Δ*csrA* strain. Data were presented either as means ± SEM. Student's *t* test was performed for statistical analysis between indicated strain and WT under indicated conditions, or between marked samples, *n* = 4 biologically independent experiments, *$p < 0.05$, **$p < 0.01$, ***$p < 0.001$.

Δ*csrA* strain reduced siderophore production to the levels below that observed in the WT strain (Fig. 3c, Supplementary Fig. 2).

CsrA typically binds to mRNAs containing GGA motif(s) in the 5′ untranslated region (5′UTR), causing changes in RNA structure, translation, stability, and/or transcription elongation[41]. To predict the CsrA-binding sites in the region upstream of the *pub* operon, the transcriptional start site was determined to be −175 by 5′RACE and the promoter (P$_{pub}$) elements such as −10 and −35 boxes were then proposed (Fig. 1d). Two potential CsrA-binding sites, AGGA that is located before ribosomal binding site (RBS) and AAGGAG that overlaps with the RBS, were identified in the 5′UTR of the *pub* transcript (Fig. 1d). These positions coincide with the findings that CsrA commonly binds to sites overlapping RBS and/or translation initiation region, competing with 30 S ribosomal subunit[41]. To probe how BarA/UvrY/Csr system regulates the expression of the *pub*, we examined the transcript levels and translation levels of *pubA* in relevant strains with qRT-PCR and the *pub*-LacZ reporter respectively. Apparently, mRNA levels of *pubA* in each of the mutants under test, including Δ*barA*, Δ*uvrY*, Δ*csrA*, Δ*barA*Δ*csrA*, Δ*uvrY*Δ*csrA*, and Δ*csrB1*Δ*csrB2*, were only slightly different from that in WT: decrease in Δ*barA*, Δ*uvrY*, and Δ*csrB1*Δ*csrB2* by about a fifth but increase in any strains lacking CsrA by about a fifth (Fig. 3d). On the contrary, the *pub*-LacZ reporter revealed that the differences in expression levels between mutants and WT were substantially more pronounced (Fig. 3d). Additionally, a vector expressing PubA with a His$_6$-tag at the C-terminus driven by the entire leader region upstream of the coding sequence (P$_{pub}$-5'UTR$_{pub}$-*pubA*) was introduced into these strains. By Western blotting, we found that the PubA levels were in excellent agreement with the *pub*-LacZ data (Fig. 3d). Consistently, repression of overexpressed CsrA on *pubA* transcription was rather modest, but became much stronger on the protein level (Supplementary Fig. 4b), supporting the proposal that CsrA inhibits *pub* expression at the post-

transcriptional level in vivo. Interestingly, it seemed that CsrB2 plays a more important role in antagonizing CsrA activity because CsrB2 was more effective than CsrB1 in elevating PubA protein levels under the same induction conditions (Supplementary Fig. 4b). Moreover, we used the constitutively active *arcA* promoter (P$_{arcA}$)[49], whose activities were comparable in these strains under experimental conditions (Supplementary Fig. 4c), in place of P$_{pub}$ to drive the expression of the His$_6$-tagged PubA and similar results were obtained (Fig. 3e). However, when 5'UTR of the *pub* operon was replaced by 5'UTR of *arcA* (P$_{arcA}$-5'UTR$_{arcA}$-*pubA*), the PubA protein levels were no longer responsive to abundance changes of any component of the BarA/UvrY/Csr system (Fig. 3e). To further support that the 5'UTR of the *pub* transcript contains the regulatory elements for CsrA, we introduced mutations into the predicted CsrA-binding motif that overlaps the RBS, including GGA-to AGA point mutation and GGA deletion. The results showed that the GGA deletion abolished expression, which can be readily explained by the removal of the RBS (Fig. 3f). In contrast, the GGA-to AGA point mutation resulted in a significant increase in expression in WT strain but a negligible change in the Δ*csrA* strain (Fig. 3f), suggesting that CrsA interacts with the *pub* transcript at the predicted CsrA-binding motif. Therefore, all of these data collectively conclude that BarA/UvrY TCS mediates expression of the *pub* operon via the pathway involving CsrB1, CsrB2, and CsrA.

## SsoR is an atypical orphan RR in terms of structure and phylogeny

We next made attempts to identify the possible cognate HK for SsoR. Information on TCSs in *S. oneidensis* was gathered from multiple sources, such as P2CS (Prokaryotic 2-Component Systems; http://www.p2cs.org)[50] and MiST3 (Microbial Signal Transduction database; https://mistdb.com)[51]. In total, the *S. oneidensis* genome encodes 103 predicted

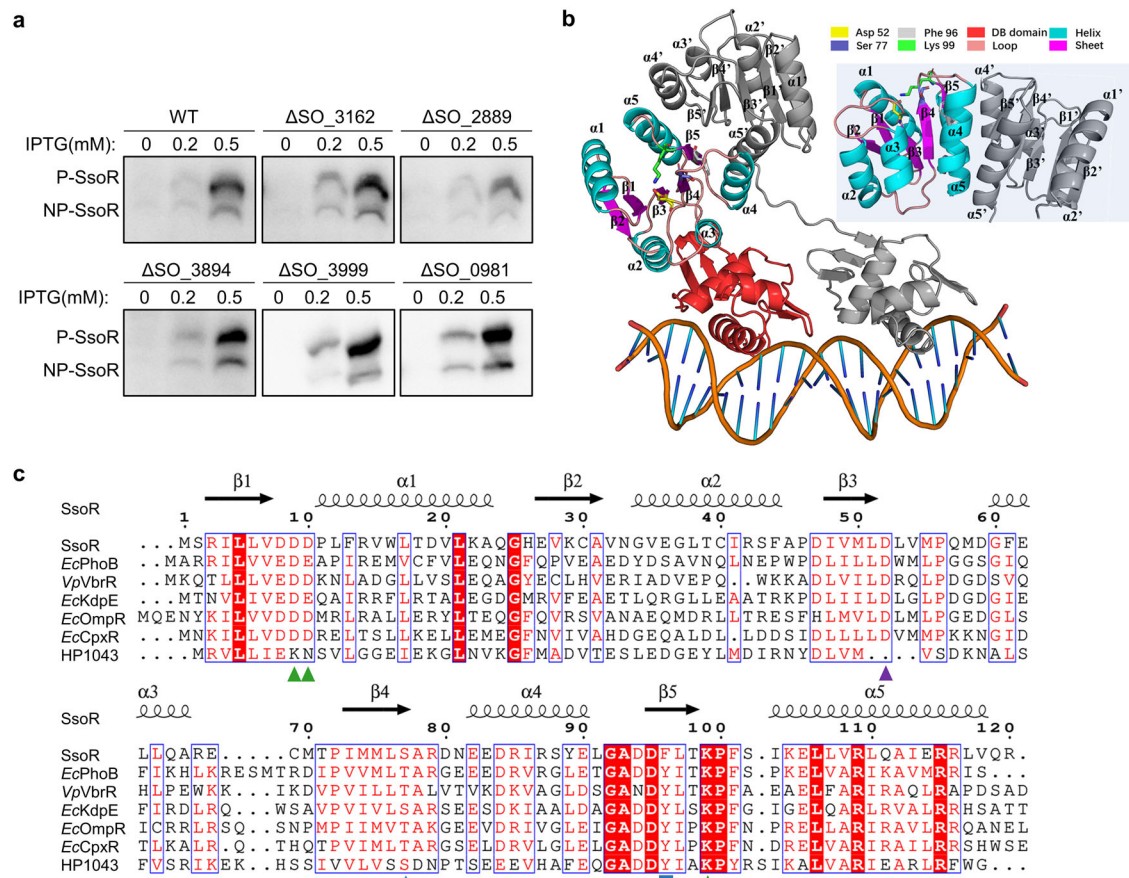

**Fig. 4 | SsoR may be an atypical orphan RR. a** Analysis of phosphorylation status of SsoR in the orphan kinases deletion mutants by Manganese (II)-Phos-tag SDS-PAGE and Western blotting. **b** Structure of SsoR. On left, the ColabFold-predicted SsoR dimer associated with DNA, refined using the "morph" function in PyMol, with *E. coli* KdpE (PDB: 4KNY) as the template. On right, the side view of SsoR's REC domain. **c** Alignment of the REC domain sequences of RRS of the OmpR/PhoB subfamily. Atypical RR (HP1043) and 5 well-characterized phosphorylation-dependent RRs are included. *Ec*, *E. coli*; Vp, *Vibrio parahaemolyticus*; HP1043, *Helicobacter pylori* HP1043. Conserved residues are boxed, and perfectly conserved residues are in red background. Secondary structures are based on SsoR.

Phosphorylation site (D52 of SsoR) is located at the α3-β3 loop pointed by brown triangle. Two highly conserved Asp/Glu residues (D8, D9 of SsoR) in the α1-β1 loop are involved in metal ion binding pointed by purple triangle). The conserved Thr/Ser (76 S of SsoR) at the end of β4 interacts with the phosphoryl group, and the subsequent small residue allows access to the phosphorylation site pointed by blue triangle. A conserved Tyr/Phe (96 F of SsoR) switch residue pointed by gray triangle in the middle of β5 and a highly conserved Lys (99 K of SsoR) residue pointed by green triangle at the end of β5 are important for phosphorylation-mediated conformational changes.

TCS components, including 43 HKs, 57 RRs, and 3 Hpts (histidine-containing phosphotransfer proteins) (Supplementary Table 1). Among them, four HK genes (SO_2889, SO_3162, SO_3894, and SO_3999) and one Hpt (SO_0981) neither are adjacent to an RR gene nor encode proteins belonging to a TCS in which the RR is experimentally confirmed (Supplementary Table 1). To test whether any of these orphan HKs could phosphorylate SsoR, we generated their in-frame deletion strains, in which His6-tagged recombinant SsoR was examined in terms of the phosphorylation status. The results showed that SsoR proteins were present in both unphosphorylated and phosphorylated forms in all the mutants as in WT (Fig. 4a), eliminating the possibility that these HKs could act as the cognate HK for SsoR.

Given that some RRs are functional in the unphosphorylated form[52,53], we then asked whether SsoR functions independent of phosphorylation. To address this, we first predicted the structure of SsoR with ColabFold and refined it based on available structures of representative members of the OmpR/PhoB subfamily. SsoR supposedly functions as a dimer, with each subunit comprising a highly conserved REC domain at the N-terminus, an unusually long flexible linker, and the C-terminal DNA-binding domain (Fig. 4b)[54–56]. While the DNA-binding domain is composed of a four-stranded β sheet, a wHTH motif, and a β hairpin, the REC domain that consists of five α-helices encircling a central β-sheet of five parallel strands,

arranged in a 21345 topology is responsible for dimerization (Fig. 4b, c)[12,57]. It has been established that the REC domain alternates between inactive and active allosteric conformations, with phosphorylation influencing this balance[56]. The phosphorylation-mediated activation depends on a common dimerization mechanism, called the Y-T coupling that involves a conserved Thr/Ser (T/S) residue (T83 and S77 in *Ec*PhoB and SsoR respectively) in the phosphorylation pocket influencing the rotameric state of a Tyr/Phe (Y/F) residue (Y102 and F96 in *Ec*PhoB and SsoR respectively) in the β5 strand, which is also called switch residue (Fig. 4b, c)[56,58]. Phosphorylation induces a conformational shift in the α4-β5-α5 face, promoting dimerization, which in turn enhances DNA binding to promoter recognition elements (Fig. 4b)[59]. Sequence and secondary structure alignments revealed that SsoR retains the conserved and essential residues of the phosphorylation pocket, including D8, D9, D52, S77, F96 and K99, unlike other characterized RRs that could be active in the non-phosphorylated form such as *Helicobacter pylori* HP1043 (Fig. 4c)[56,60].

We then carried out the analyses of sequences and evolutionary relationships of representative RRs, including SsoR. The Uniref50 sequences of all OmpR subfamily RRs were retrieved, and five clusters including PhoBs, SsoRs, VbrRs, KdpEs, and CusRs were selected to construct an evolutionary tree for structure alignments (Fig. 5a). For each cluster, all members have the same genomic backgrounds, either standing alone or next to an HK gene

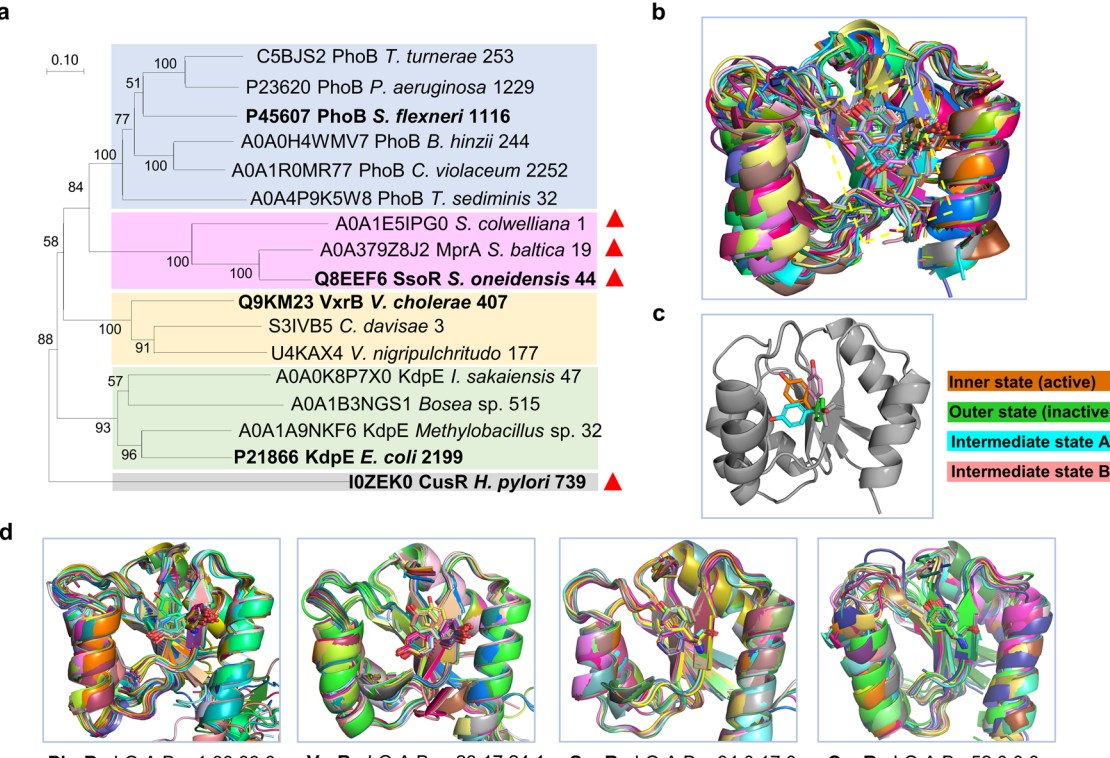

**Fig. 5 | SsoR likely represents a unique group of phosphorylation-independent RRs. a** A phylogenetic tree of representative RRs of the OmpR/PhoB subfamily. Five clusters shown by different background colors and named by the proteins in bold, contain homologous proteins from the Uniref50 clusters in Uni-ProtKB. For each clade, the UniProt ID, gene name, species, and the number of enclosed members are given. Orphan RRs were marked with red triangles. Full names of bacteria in the tree and the genomic context of the evolutionary tree members are detailed in the Supplementary Fig. 5. **b** The alignment result of the REC domains of 69 RRs obtained from PDB, with the conformational states of the switch residues in yellow box being inner state: outer state: intermediate state A: intermediate state B = 42:19:5:1 (two structures belonging to an intermediate state situated between the inner state and intermediate state A). All proteins are shown as cartoons, while F/Y switches are depicted as sticks, and the α4-helix is hidden for better visualization. **c** A more intuitive schematic diagram of four states, depicting the protein as a cartoon, with the switch residues displayed as sticks in various colors to represent different orientations. **d** From left to right, four diagrams are alignment results of the AlphaFold2-predicted structures of PhoBs, SsoRs, VbrRs, and CusRs clusters, respectively. The occurrences of four states were counted.

(Supplementary Fig. 5). Apparently, the SsoRs cluster is small, compared to those made of other RRs (Fig. 5a). Intriguingly, the SsoRs cluster seemed to have emerged at the same time as the PhoBs cluster during evolution, but the VbrRs and KdpEs clusters diverged more anciently (Fig. 5a). The CusRs cluster, composed of the homologs of *H. pylori* HP1043, separated from other RRs in the tree even earlier (Fig. 5a).

In the OmpR/PhoB subfamily RRs, the switch residue (Fig. 4b, c) could work as an indicator of an RR's status[56,61,62]. Indeed, the REC domains of the OmpR/PhoB subfamily RRs from the PDB database share highly similar structures but vary in the orientation of the switch residues (Fig. 5b, c). To further verify this, we generated a structural similarity dendrogram with the *Ec*PhoB PDB structures and predicted structures as AlphaFold2 can generate various conformations that exist naturally, even with identical input protein sequences, which are equivalent to multiple same sequences[63]. The REC domain structures of 55 proteins sharing identical sequences with *Ec*PhoB were collected from AlphaFold Protein Structure Database for dendrogram construction using DALI (Supplementary Fig. 6a), which summarizes the occurrence frequency of all possible conformations, that is, the orientation of the switch residue (Fig. 5c). The switch residues in the predicted structures were found to be in one of four major different orientations, inner (active, phosphorylated), outer (inactive, non-phosphorylated), and two intermediate states, which are less frequent under native conditions (Fig. 5c). Importantly, from identical PhoB proteins, the conformations isolated by the orientation of the switch residues tend to cluster into distinct groups on the dendrogram, supporting that the predicted structures are consistent with those obtained experimentally (Supplementary Fig. 6a–c).

Then, the predicted REC domain structures of the members in the evolutionary tree were gathered from the AlphaFold Protein Structure Database, aligned in PyMol, and the numbers of all varying states were counted (Fig. 5c). The switch residues in two groups of phosphorylation-dependent RRs, PhoBs and VbrRs, could be active, inactive, and intermediate states. In contrast, the proportion of switch residue orientations in the SsoRs and CusRs clusters notably differs from that of the PhoBs and VbrRs clusters. Specifically, the switch residue of the phosphorylation-independent CusRs cluster exclusively exhibits inward orientations (Fig. 5d), consistent with observations in the structure of the CusRs cluster member HP1043 (PDB: 2PLN)[60], and a similar scenario was found with the SsoRs cluster members. More importantly, the lack of the occurrence of outer state in these two groups of RRs indicates that they could not exist in unphosphorylated inactive form (Fig. 5d), strongly supporting that SsoR possibly is active independent of phosphorylation.

## SsoR regulates transcription in a phosphorylation-independent manner

To address that SsoR probably employs a phosphorylation-independent activation mechanism, we compared the regulation activity of two SsoR variants carrying mutations at the phosphorylation residue (D52), SsoR[D52N] and SsoR[D52E]. Both variants are in the non-phosphorylated form, but the Asp to Glu mutation is phosphomimetic[64], that is, SsoR[D52E] would be constitutively active. Unlike SsoR, both SsoR[D52N] and SsoR[D52E] migrated on SDS-PAGE as a single band independent of Phos-tag (Fig. 6a), validating that they exist in the non-phosphorylated form only. To assess the regulatory activity, these SsoR variants were expressed in the Δ*ssoR* strain and

**Fig. 6 | SsoR regulates the *pub* operon independent of phosphorylation. a** Conventional SDS-PAGE (without Mn(II)-Phos-tag), Mn(II)-Phos-tag SDS-PAGE, and Western blotting of WT and mutant SsoR proteins. C-terminus His$_6$-tagged SsoR variants were induced with 0.2 and 0.5 mM IPTG in indicated strains. **b** Siderophore production of Δ*ssoR* expressing SsoR variants. **c** Expression of *pub* of Δ*ssoR* expressing SsoR variants by the *pub*-LacZ reporter. **d** EMSA analysis of SsoR variants with *pub* promoter sequence. His$_6$-tagged SsoR variants, expressed and purified from *E. coli*, were mixed with the biotin-tagged *pub* promoter DNA of ~300 bp. EMSA assay was performed with 40 nmol biotin-labeled *pub* promoter and various amounts of SsoR proteins (0, 20, 40, 80, 160, 200 nmol). Non-specific competitor DNA (40 nmol poly(dI·dC) was included in all lanes. Promoter fragment of the 16 s rRNA gene (P$_{16S}$) was used as the negative control. Data were presented either as means ± SEM. Student's *t* test was performed for statistical analysis between indicated strain and WT under indicated conditions, or between marked samples, *n* = 4 biologically independent experiments, \**p* < 0.05, \*\**p* < 0.01, \*\*\**p* < 0.001.

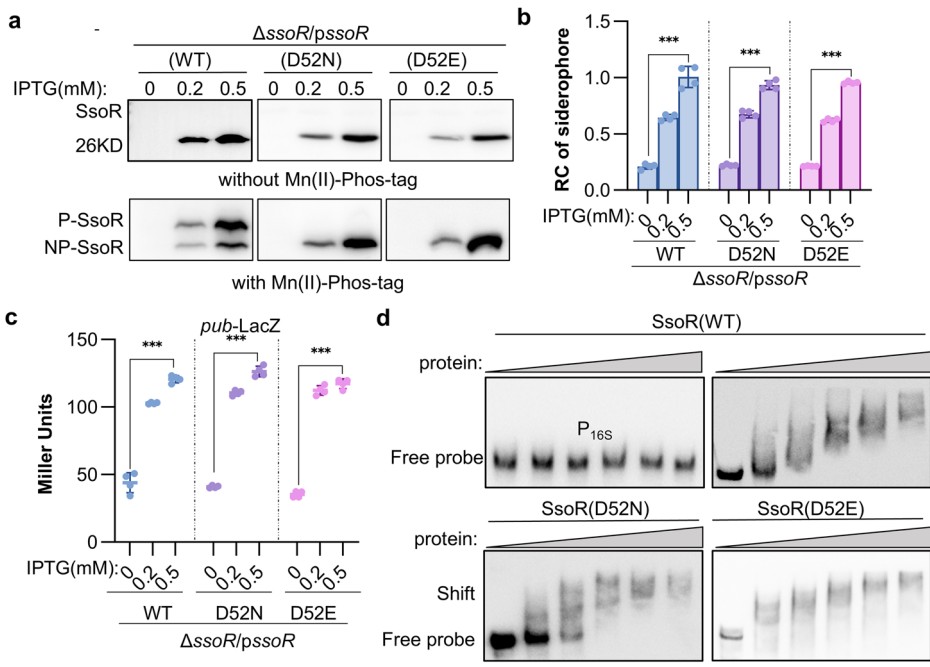

siderophore production and *pub* expression were examined. When expressed at the same levels, both SsoR$^{D52N}$ and SsoR$^{D52E}$ behaved indistinguishably from SsoR (Fig. 6b, c, Supplementary Fig. 2), indicating that they are functional. In addition, His$_6$-tagged recombinant SsoR, SsoR$^{D52N}$, and SsoR$^{D52E}$ proteins were expressed and purified from *E. coli* (Supplementary Fig. 7), and direct interaction between the purified proteins and the *pub* promoter region were analyzed by EMSA. Apparently, all SsoR variants bound well to the DNA fragment in comparison with negative control P$_{16s}$ (the promoter sequence of the 16 s rRNA gene) (Fig. 6d). Altogether, these data allow us to conclude that SsoR directly regulates transcription of the *pub* operon in a phosphorylation-independent manner.

## Conformational occurrence of the switch residue in SsoR

To unravel the mechanism underlying the phosphorylation-independent regulation of SsoR, comparative analyses of SsoR and phosphorylation-dependent *Vp*VbrR (VbrR from *Vibrio parahaemolyticus*) were conducted with all-atoms molecular dynamics (MD) simulations. The systems were built on the phosphorylation mimic VbrR$^{D-RD-D51E}$ (PDB: 7E90) dimer, the non-phosphorylated VbrR$^{D-RD-D51N}$ dimer, and the monomers (VbrR$^{M-RD-D51N}$ and VbrR$^{M-RD-D51E}$) extracted[65]. In parallel, we constructed systems for SsoR$^{D-RD-D52E}$, SsoR$^{D-RD-D52N}$, SsoR$^{M-RD-D52E}$, and SsoR$^{M-RD-D52N}$. Each system underwent a 3 μs MD simulation, and the resulting trajectories were generated and analyzed, each consisting of 3000 frames (1 ns per frame).

According to the Chi1 angle of the switch residue and the distance between the backbone N of SsoR$^{98T}$ or VbrR$^{99T}$ and the CZ atom of SsoR$^{96F}$ or VbrR$^{97Y}$ switch residue, the frames were categorized into four dynamics states as described in Fig. 5c (Supplementary Movie 1). In fact, the conformational changes among "inner state", "intermediate state A", and "intermediate state B" are continuous processes, encompassing a large number of intermediate states. A statistical analysis of the distribution of the four states in the trajectories was then conducted (Fig. 7a-h). It was observed that the outer state only exists in VbrR$^{D-RD-D51N}$ and VbrR$^{M-RD-D51N}$ (Fig. 7f, h) but not VbrR$^{D-RD-D51E}$ and VbrR$^{M-RD-D51E}$ (Fig. 7e, g), indicating that the mimicked phosphorylation in phosphorylation site blocks conformational transition from inner state to outer site. The distribution patterns of these states differ between the monomeric and dimeric forms. The intermediate state B is notably more prominent in VbrR$^{M-RD-D51E}$ (Fig. 7d) compared to VbrR$^{M-RD-D51N}$ (Fig. 7h), but the trend is opposite in dimers (Fig. 7b, f).

Moreover, in the dimeric form, a higher occurrence of intermediate state A is observed in VbrR$^{D-RD-D51N}$ (Fig. 7f). Furthermore, the distribution ratio of inner and outer states is notably higher in VbrR$^{D-RD-D51N}$ (Fig. 7f) than that of VbrR$^{M-RD-D51N}$ (Fig. 7h), demonstrating that a tendency exists in the inner state when in the dimeric configuration. In addition, it is worth mentioning that the MD simulation results of VbrR$^{M-RD-D51N}$ are consistent with the alignment results of VbrRs (Figs. 5d and 7f).

The conformational occurrences of the switch residues in SsoR$^{D-RD-D52E}$ (Fig. 7a), SsoR$^{M-RD-D52E}$ (Fig. 7c), SsoR$^{D-RD-D52N}$ (Fig. 7e), and SsoR$^{M-RD-D52N}$ (Fig. 7g) are notably uniform, with the inner state being predominant, and only a small number of the outer states being observed in SsoR$^{M-RD-D52N}$ (Fig. 7g). Although the occurrences of the inner state between SsoR$^{M-RD-D52E}$ (Fig. 7c) and SsoR$^{M-RD-D52N}$ (Fig. 7g) exhibit some differences as the switch residue is more stable in SsoR$^{M-RD-D52E}$ (RMSF of switch residue = 0.111 in SsoR$^{M-RD-D52E}$, 0.136 in SsoR$^{M-RD-D52N}$) and outer state appears in SsoR$^{M-RD-D52N}$ but not SsoR$^{M-RD-D52E}$ (Fig. 7c, g), the MD simulations overwhelmingly support that the state of the phosphorylation site has almost no effect on SsoR. By combining physiological and biochemical data presented above, we conclude that SsoR represents a group of OmpR/PhoB subfamily RRs that are unique in that they function in a phosphorylation-independent manner despite retaining a phosphorylation pocket and the popular 'Y-T coupling' mechanism.

## Fur acts as an iron sensor to regulate transcription of the *pub* operon both directly and via SsoR

A Fur-binding site (gcagatgagaacgatttgc, −210 to −192 relative to the start codon) partially overlapping the −35 box of the *pub* promoter was reported before[66], implying that Fur likely acts as a transcriptional repressor for the *pub* operon (Fig. 1e). By using EMSA, we first substantiated the direct interaction between purified His$_6$-tagged Fur and the *pub* promoter sequence (Fig. 8a). Then the repressing effect of Fur on transcription of the *pub* operon was confirmed by qRT-PCR and Western blot: the Fur loss resulted in substantially increased transcription, and this elevation was no longer responsive to the changes in iron levels (Fig. 8b). In addition, we found that Fur is also responsible for sensing iron in the absence of SsoR with a *fur ssoR* double mutant, in which the *pub* expression was found not to be altered significantly upon changes in iron levels (Fig. 8b).

As a phosphorylation-independent transcriptional regulator, SsoR likely enhances expression of its target genes by increasing its own

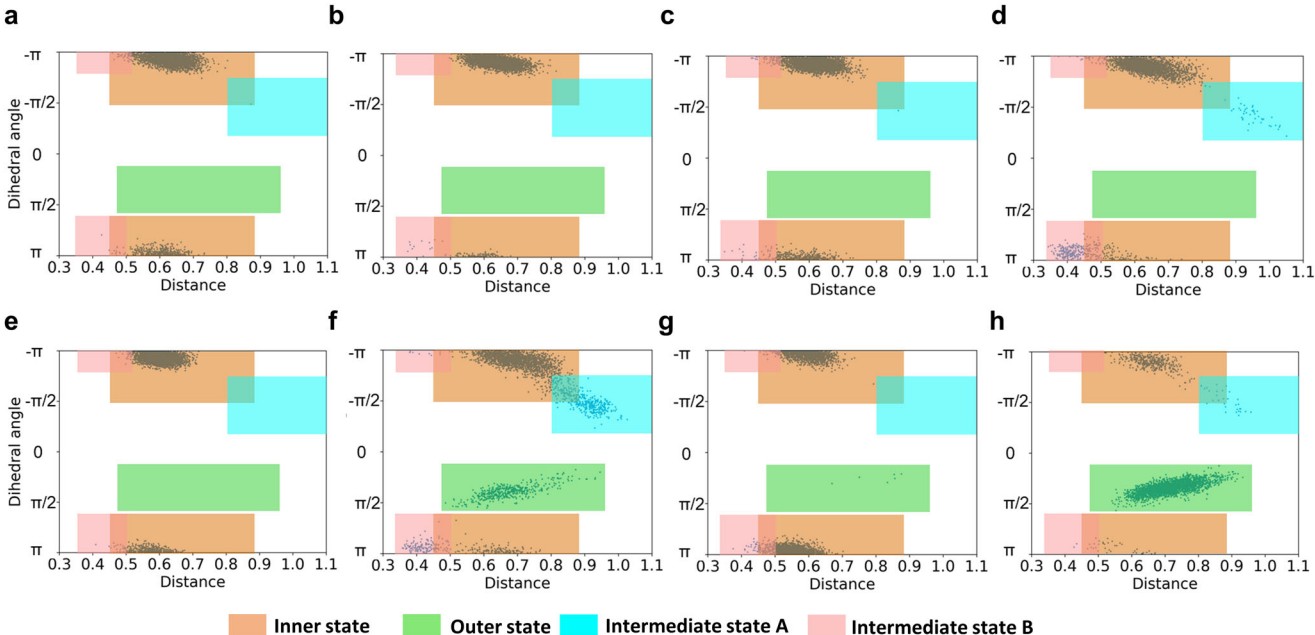

**Fig. 7 | Statistical analysis of the occurrence of four states of the switch residue of SsoRs and VbrRs in the trajectories.** The MD simulations generated trajectories of SsoRs and VbrRs consisting of 3000 frames (1 ns per frame). The state occurrence of the switch residue of SsoRs and VbrRs was clustered into four major states, in orange, green, light blue, light red backgrounds. The horizontal coordinate indicates the distance between the CZ atom of 96 F(SsoR)/97Y(VbrR) and the backbone N atom of 98 T(SsoR)/99 T(VbrR). The vertical coordinate represents Chi1 angle of 96 F(SsoR)/97Y(VbrR). The analysis of relative positions of 96 F(SsoR)/97Y(VbrR) using Plumed, RMSF was calculated in GROMACS. **a** SsoR$^{D-RD-D52E}$, **b** VbrR$^{D-RD-D51E}$, **c** SsoR$^{M-RD-D52E}$, **d** VbrR$^{M-RD-D51E}$, **e** SsoR$^{D-RD-D52N}$, **f** VbrR$^{D-RD-D51N}$, **g** SsoR$^{M-RD-D52N}$, **h** VbrR$^{M-RD-D51N}$.

abundance. Our previous proteomic analyses revealed a 4.32-fold increase in SsoR protein levels in the Δ*fur* strain compared to WT, suggesting that Fur represses SsoR expression[67]. In line with this, a Fur-binding motif was identified in the *ssoR* promoter region, which partially overlaps with the −10 box and the transcription start site (Fig. 8c). With LacZ reporter and Western blotting, we found that the expression of *ssoR* was induced inversely proportional to iron levels in the WT strain but became constitutive at significantly higher levels in the Δ*fur* strain (Fig. 8d). Moreover, SsoR is subject to self-regulation. The *ssoR*-LacZ reporter assay showed that the absence of *ssoR* enhanced β-galactosidase activity considerably compared to that of the WT strain grown under the same conditions (Fig. 8d). Although the additional removal of Fur abolished the response to iron levels, the repressing effect of SsoR on its own expression was still observable (Fig. 8d). Furthermore, we substantiated that SsoR proteins in either phosphorylated or non-phosphorylated form were able to bind with the *ssoR* promoter fragment (Fig. 8e), strengthening that SsoR functions independently of phosphorylation. All these data indicate that in *S. oneidensis*, Fur is the primary, if not exclusive, iron sensor and, by sensing changes in intracellular iron levels, influences siderophore biosynthesis both directly and indirectly. In cells grown under iron-repleted conditions, Fur is sufficient to repress the transcription of the *pub* operon, but when iron is scarce, Fur-mediated repression is relieved, and transcription is activated by SsoR. We envision that self-regulation of SsoR offers an additional safeguarding mechanism to prevent this activity-unconstrained regulator from overproduction.

## Discussion

*Shewanella* are found in a wide range of ecological niches and play a critical role in global element cycles because of their unparallel respiration versatility. This capacity is largely based on iron proteins, and, therefore *Shewanella* usually has high iron demand, which relies on multiple strategies for iron uptake[16–18]. One of the unique features of most *Shewanella* is the presence of a single enzyme system for biosynthesis of an array of siderophores[8,23]. Importantly, some of the siderophores have additional

activities, such as inhibition of motility and biofilm formation, and would conceivably have a profound ecological impact on shaping local community[7,8]. However, our understanding of the regulatory mechanisms behind siderophore synthesis in *Shewanella* is still limited. In this work, we identified two TCSs, along with Fur, that modulate the siderophore production at multiple levels. While BarA/UvrY relies on an sRNA-dependent cascade, SsoR is an RR that functions in a phosphorylation-independent manner (Fig. 9).

Identification of these two regulators was enabled by the unexpected color-loss phenotype of the siderophore-overproducing strain Δ*putA*[27,28]. Disruption of either *barA* or *ssoR* by transposon insertion compromises siderophore production, restoring the signature colony color. Although it is attractive to speculate that BarA and SsoR may belong to the same regulatory pathway, BarA does not affect the phosphorylation state of SsoR in vivo. Instead, BarA constitutes a TCS with UvrY as in many other bacteria hosting this system[30,32], affecting the expression of the siderophore biosynthesis system PubABC via the post-transcriptional regulatory mechanism (Fig. 9). Our study shows that the BarA/UvrY system of *S. oneidensis*, in line with its counterparts in other γ-proteobacteria such as *E. coli* and *P. aeruginosa*, employs Csr/Rsm cascade to regulate siderophore synthesis[9,68]. At least two sRNAs, CsrB1 and CsrB2, and RNA chaperone CsrA were identified to play critical roles in transducing signals perceived by BarA and relayed by UvrY to control siderophore biosynthesis. Multiple lines of evidence were presented to support that CsrA directly interacts with the *pub* transcript to block translation. This effect is antagonized by CsrB1 and CsrB2, whose transcription is activated by UvrY upon phosphorylation. However, given that the BarA/UvrY/Csr regulatory network is rather complex, featuring autoregulatory circuitry and the involvement of various factors like cAMP-CRP and RpoE[42,69,70], further investigation is needed to identify other factors that influence siderophore synthesis through the BarA/UvrY/Csr pathway in *Shewanella*. Since the physiological stimulus for BarA has been suggested to be metabolic end products[39,42,69,70], we speculate that perhaps shifts in carbon metabolism or some secondary metabolite processes trigger the response, thereby putting siderophore biosynthesis

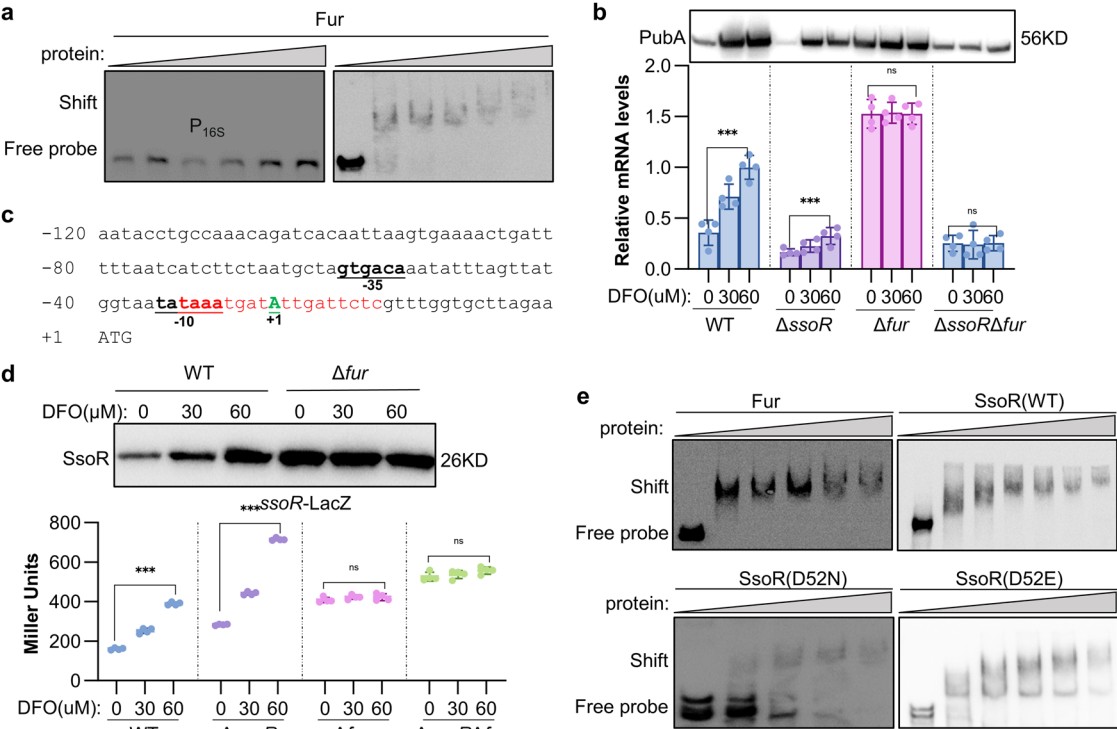

**Fig. 8 | Fur acts as an iron sensor to regulate transcription of the *pub* operon both directly and via SsoR. a** EMSA assay was performed with 40 nmol biotin-labeled *pub* promoter and various amounts of Fur protein (0, 20, 40, 80, 160, 200 nmol). Non-specific competitor DNA (40 nmol poly(dI·dC) was included in all lanes. **b** Expression of *pub* in indicated strains under different iron level conditions. Protein and mRNA levels of *pubA* were assayed by western blot and qRT-PCR, respectively. His$_6$-tagged PubA was expressed by the *pub* native promoter. **c** Schematic diagram of the *ssoR* promoter region. The Fur-binding motif is in red. The predicted −10 box and −35 box promoter regions are underlined. +1 represents the transcription start site. **d** Expression of *ssoR* in indicated strains under different iron conditions by western blotting and *ssoR*-LacZ reporter. His$_6$-tagged SsoR was expressed by the *ssoR* native promoter. **e** EMSA assay was performed with 40 nmol biotin-labeled *ssoR* promoter and various amounts of Fur and SsoR proteins (0, 20, 40, 80, 160, 200 nmol). Non-specific competitor DNA (40 nmol poly(dI·dC) was included in all lanes. Data were presented either as means ± SEM. Student's *t* test was performed for statistical analysis between indicated strain and WT under indicated conditions, or between marked samples, *n* = 4 biologically independent experiments, *$p < 0.05$, **$p < 0.01$, ***$p < 0.001$.

**Fig. 9 | A model for the regulation by BarA/UvrY and SsoR TCSs and Fur of siderophore synthesis.** In this model, BarA senses metabolic cues to phosphorylate UvrY, which in turn activates the transcription of sRNA CsrB1 and CsrB2. CsrB1 and CsrB2 antagonize CsrA to relieve the translational repression of the *pub* mRNA. CsrA represses the translation of *pub* by binding to the 5'UTR. Transcriptional regulation of *pub* by SsoR and Fur is iron-responsive. Under iron-repleted conditions, Fur binds to the RNA polymerase binding site in the promoter region of *ssoR* gene and *pub* operon to repress the transcription. Under iron-depleted conditions, Fur falls off from the *ssoR* and *pub* promoter regions, resulting in transcriptional derepression. Then, the increased SsoR binds to the *pub* promoter to upregulate transcription in a phosphorylation-dependent manner.

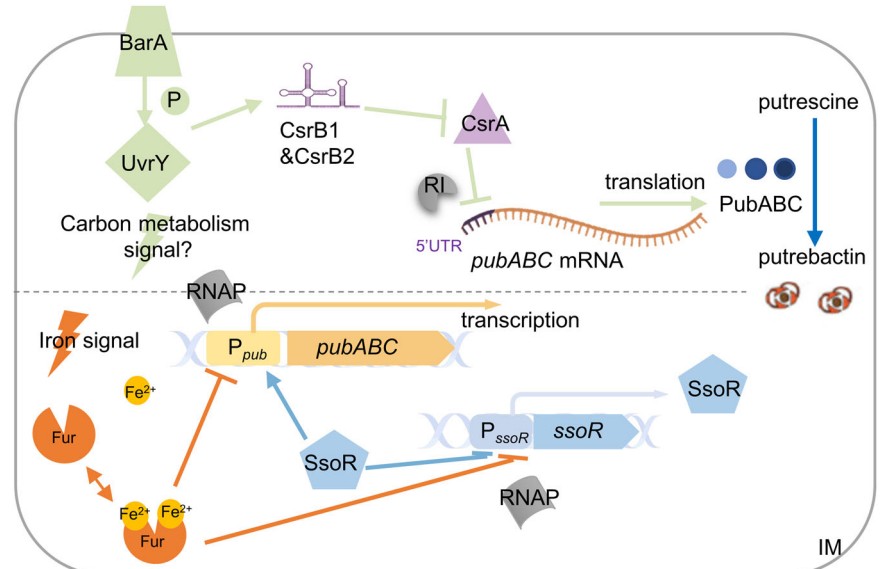

under the control of cellular metabolic status. Coupling carbon metabolism and iron uptake may be particularly important for *Shewanella* as this group of bacteria encodes a vast number of iron-containing proteins, many of which are involved in metabolism as enzymes and electron carriers[71,72].

RRs of the OmpR/PhoB subfamily are thought to become activated through phosphorylation, which triggers an allosteric change to enable homodimerization and enhance DNA binding[56,59]. However, some are able to bind DNA without phosphorylation to exert different functions not observed from their phosphorylated counterparts[52,53,73]. Hence, one of the

most striking findings in this study is that as an RR of the OmpR/PhoB subfamily, SsoR regulates siderophore synthesis in a phosphorylation-independent manner. By examining the phosphorylation state of SsoR in vivo, we eliminated the possibility that the orphan HKs that phosphorylate SsoR are present. Results of both in vivo gene expression analysis and in vitro EMSA assay support that regulation of the *pub* operon transcription by SsoR is not dependent on phosphorylation, providing a case for non-canonical functioning modes of the OmpR/PhoB subfamily.

In contrast to atypical RRs reported before that do not require phosphorylation for activity[60,64], SsoR has key conserved sites for phosphorylation-dependent regulation, and in the evolutionary tree, the SsoRs cluster is closely related to the typical PhoB RRs. By aligning AlpahFold2 predicted structures, we found distinctive behaviors of the F/Y switch residue of SsoRs from those of the phosphorylation-dependent RRs, such as PhoBs and VbrRs. Importantly, MD simulations further reveal the contrasting effects of phosphorylation on the conformational occurrences of the switch residues in typical VbrRs and atypical SsoRs. The switch residue in phosphorylation-dependent RRs could be in any state, but its counterpart in SsoR is locked in the active inner state only. Nevertheless, SsoRs belong to the OmpR/PhoB subfamily as they retain all conserved features observed from the phosphorylation-dependent members, the 'Y-T coupling' mechanism in particular[56]. This is in sharp contrast to CusRs (HP1043), whose independence of phosphorylation is due to the lack of the phosphorylation residue (Fig. 4c). Thus, SsoRs represent a unique group of OmpR/PhoB subfamily RRs that evolve out a phosphorylation-independent activating mechanism from the conventional phosphorylation-dependent chassis. How this occurs can be addressed by more in-depth structural analysis and MD simulations, which are underway.

Although phosphorylation is not required for functionality, SsoR retains the ability to be phosphorylated because of a highly conserved phosphorylation pocket[56,74]. As a result, a portion of SsoR appears to be constitutively phosphorylated in the cell, indicating the presence of phosphate donors. Given that none of the orphan HKs are found to be the exclusive cognate HK for SsoR, candidate phosphate donors should be alternative orphan kinases, non-cognate kinases, and/or small-molecule high-energy phosphodonors, such as phosphoramidate and acetyl phosphate[75,76].

In addition to BarA/UvrY TCS and SsoR, Fur regulates the expression of the *pub* operon both directly and indirectly in response to iron availability. Under iron-repleted conditions, Fur binds to the Fur-boxes in the promoter region of the *ssoR* gene and the *pub* operon, which overlap the RNA polymerase binding site, to repress transcription. Under iron-depleted conditions, Fur falls off, allowing transcription of both *pubABC* and *ssoR*. SsoR seems to be produced in needed quantity upon the Fur removal, which in turn provides additional activation for *pubABC* transcription. In addition, SsoR represses its own expression, preventing overproduction. Clearly, only when SsoR and Fur work together, *S. oneidensis* cells are capable of rapidly upregulating the siderophore synthesis when faced with iron-depleted condition.

Some atypical phosphorylation-independent RRs which lack HK adopt alternative strategies, such as post-translational acetylation, to regulate their own activity[56]. Our research has revealed a unique paradigm, SsoR is not phosphorylation-dependent but dose-dependent to regulate the transcription of its regulon. Here, Fur acts as the sensor for SsoR. Fur's regulation and self-regulation together prevent SsoR from being constitutively active. We suggest that perhaps accidental loss of the HK gene occurred first, forcing SsoR to select another type of sensory partner and evolve a phosphorylation-independent activation capacity, or perhaps Fur regulation occurred first, leading to redundancy of HK. Either way, this merits further investigation. Overall, this study suggests that through the orchestrated regulatory network, different signals, i.e., iron availability or central metabolic state, are integrated into the multilayered regulation of siderophore synthesis, providing more insights into the current understanding of already complex regulatory mechanisms for siderophore production in bacteria.

## Methods

### Bacterial strains, plasmids, and culture conditions

Bacterial strains and plasmids used in this study are listed in Supplementary Table 2. Information for primers used in this study is given in Supplementary Table 3. Chemicals were obtained from Sigma-Aldrich (Shanghai, China) unless otherwise noted. *E. coli* and *S. oneidensis* strains were grown under aerobic conditions in Lennox LB (Difco, Beijing, China) under aerobic conditions at 37 and 30 °C for genetic manipulation. When needed, the growth medium was supplemented with chemicals at the following concentrations: 2,6-diaminopimelic acid (DAP), 0.3 mM; ampicillin sodium, 50 µg/ml; kanamycin sulfate, 50 µg/ml; and gentamycin sulfate; 15 µg/ml.

### In-frame mutant construction and genetic complementation

In-frame deletion strains for *S. oneidensis* were constructed using the *att*-based fusion PCR method[77]. In brief, two fragments flanking the gene of interest were amplified and then joined together by a second round of PCR. The resulting fusion fragment was introduced into suicide plasmid pHGM01 by site-specific recombination using the BP Clonase (Invitrogen, Carlsbad, CA) and the resulting mutagenesis vectors were maintained in *E. coli* DAP-auxotroph WM3064. The vectors were then transferred from *E. coli* into the relevant *S. oneidensis* strain by conjugation. Integration of the mutagenesis construct into the chromosome was selected by gentamycin resistance and confirmed by PCR.

For genetic complementation of the mutants and inducible gene expression, genes of interest generated by PCR were cloned into pHGEN-Ptac under the control of IPTG-inducible promoter $P_{tac}$[78]. After verification by sequencing, the resultant vectors in *E. coli* WM3064 were transferred into the relevant strains via conjugation.

### Site-directed mutagenesis

Site-directed mutagenesis was performed to generate SsoR proteins carrying point mutations (D52N and D52E) using a QuikChange II XL site-directed mutagenesis kit (Agilent, Beijing, China). The *ssoR* gene within pHGEN-Ptac and pET-28a(+) was subjected to modification, and the resulting products were digested by DpnI at 37 °C for 6 h and subsequently transformed into *E. coli* WM3064. The vectors carrying the intended mutations, which were verified by sequencing, were transferred into the relevant *S. oneidensis* and *E. coli* strains by conjugation.

### Transposon mutagenesis

A random mutation library for the ΔputA strain, which forms white colonies on LB agar plates, was constructed with mariner-based plasmid pFAC[79,80]. A total of ~15,000 random mutants were screened for reddish-brown colonies on LB agar plates supplemented with gentamycin. To identify the transposon insertion sites in these isolates, arbitrary PCR was employed[81].

### Heme *c* assays

Cultures of *S. oneidensis* strains grown in liquid LB to the early stationary phase were centrifuged, and the pellets were photographed. The cytochrome *c* abundance of strains was first estimated by the color intensity of the cell pellets. Subsequently, the pellets were suspended in phosphate-buffered saline (PBS, pH 7.0), adjusted to the same $OD_{600}$ values, and the cells from the same-volume aliquots were disrupted. All proteins were precipitated by trichloroacetic acid precipitation[82] and assayed for heme *c* levels with the QuantiChrom heme assay kit (BioAssay Systems, CA, USA) according to the manufacturer's instructions.

### Siderophore measurement

To visualize siderophores, *S. oneidensis* strains grown on LB agar plates were subjected to Chrome Azurol S (CAS) plate assay using CAS and Hexadecyltrimethylammonium bromide (HDTMA) as indicators. Siderophores with higher iron affinity scavenge iron from the Fe-CAS-HDTMA complex, and subsequent release of the CAS dye results in a color shift from

blue to orange[83]. Ten microliters of cultures of the mid-exponential phase ($OD_{600}$, ~0.6, the same throughout the study) were dropped and incubated on LB agar plates containing 30 mM DFO for 24 h, followed by pouring in CAS reagent to completely cover the entire plate. The formation of chelated halos was observed and photographed three hours later. To quantify total siderophores, *S. oneidensis* strains were grown in liquid LB to the stationary phase, and cell-free culture supernatants were obtained by centrifugation. Siderophore concentrations within the supernatants were determined using the liquid CAS assay[83].

### SDS-PAGE, Mn(II)-Phos-tag SDS-PAGE, and western blotting assays

Cells entering the mid-exponential growth phase were harvested by centrifugation, washed with Tris/HCl (pH 7.0) buffer containing phosphatase inhibitors (Solarbio, Beijing, China), resuspended in the same buffer, and sonicated. Throughout this study, the total protein concentration of the cell lysates was determined by the bicinchoninic acid assay using bovine serum albumin (BSA) as a standard or using a GE NanoVue Spectrophotometer for fast assessment. Conventional SDS-PAGE was performed using slab gels consisting of a 10% acrylamide separating gel, and a 5% stacking gel. Mn(II)-Phos-tag SDS-PAGE was used to separate SsoR and UvrY proteins in different phosphorylation states. Fifty μM acrylamide-pendant Phos-tag ligand (Wako Pure Chemical, Osaka, Japan) and 100 μM $MnCl_2$ were added to a 10% separating gel before polymerization according to the instructions provided by the Phos-tag Consortium[84]. After electrophoresis, Phos-tag acrylamide gels were washed with transfer buffer (50 mM Tris, 384 mM glycine, 20% methanol) containing 1 mM EDTA for 10 min with gentle shaking and then with transfer buffer without EDTA for 10 min to remove $Mn^{2+}$.

Proteins on the PAGE gels were then electrophoretically transferred to PVDF membrane (Millipore, Bedford, MA) according to the manufacturer's instructions (Bio-Rad, Hercules, CA, USA). Tris Buffered Saline with 0.1% Tween containing 5% BSA was used to block the membrane. The membrane was probed with a 1:5000 dilution of a mouse monoclonal his-tag antibody (Abbkine, Shanghai, China), followed by a 1:10,000 dilution of Goat anti-mouse IgG-HRP (horseradish peroxidase) (Beyotime, Beijing, China) and the signal was detected using a chemiluminescence Western blotting kit (Roche, Basel, Switzerland). Images were visualized with Che-miScope 6000 Imaging System (Clinx, Shanghai, China).

### LacZ reporter assay

Expression of target genes was assessed using a single-copy integrative LacZ reporter system[85]. Briefly, fragments containing the sequence upstream of the target operons (−500 to +1 relative to the translation start codon) were amplified, cloned into the reporter vector pHGEI01, and transformed into *E. coli* WM3064 and verified by sequencing. The correct vector was then transferred by conjugation into relevant *S. oneidensis* strains, which it integrated into the chromosome. Cells of the mid-exponential phase under test conditions were harvested by centrifugation, washed with PBS, and lyzed with the lysis buffer (0.25 M Tris/HCl, pH 7.5, 0.5% Triton X-100). The resulting soluble protein was collected after centrifugation and used for enzyme assay by adding the aliquot of the o-nitrophenyl-β-D-galactopyr-anoside (4 mg/ml). β-galactosidase activity was determined by monitoring color development at 420 nm using a Synergy 2 Pro200 Multi-Detection Microplate Reader (Tecan, Männedorf, Switzerland), and results were presented as Miller units.

### Quantitative RT-PCR (qRT-PCR)

Total RNAs were extracted using a Trizol reagent (Invitrogen, Carlsbad, CA, USA) following the manufacturer's instructions. The extracted RNAs were purified using an RNeasy Mini Kit and RNase-Free DNase Set (Qiagen, Valencia, CA, USA). The QuantiTect Reverse Transcription Kit (Qiagen, Valencia, CA, USA) was used to synthesize cDNA.RT-qPCR was performed using 2xSYBR Green PCR Mastermix (Solarbio, Beijing, China) and

monitored in CFX Opus Real-time PCR System (Bio-Rad, Hercules, CA, USA). The cycle threshold (CT) values for each gene of interest were averaged and normalized against the CT value of the *arcA* gene, whose abundance was constant during the exponential phase. The relative abundance (RA) of each gene compared with that of *arcA* was calculated using the $2^{-\Delta\Delta CT}$ method[86]. The expression of each gene was determined from four biological replicates, and in a single qRT-PCR experiment, three replicates were measured.

### Detection of protein levels in vivo

The fragments containing the natural or recombinant leader region and open reading frame of *pubA* and *ssoR* genes with $His_6$-tag at the C-terminus were amplified, cloned into the promoterless and low-copy plasmid pHG101[87], and transformed into *E. coli* WM3064 and verified by sequencing. The correct vector was then transferred by conjugation into relevant *S. oneidensis* strains. To detect protein levels in vivo, cells were cultured under relevant conditions, and proteins were extracted and subjected to SDS-PAGE and Western blotting.

### Recombinant protein expression and purification and EMSA

*E. coli* BL21(DE3) and the pET-28a(+) plasmid were used for the production of recombinant SsoR and Fur with $His_6$-tag at the N-terminus[66]. Expression of SsoR and Fur in *E. coli* BL21 cells was induced with 0.2 mM IPTG from the mid-exponential phase at 16 °C overnight. The cells were grown to saturation and then collected by centrifugation resuspended in lysis buffer (50 mM Tris/HCl, pH 7.5, 500 mM NaCl, 1 mM PMSF, 5 μg/ml DNaseI), and broken by passage twice through a French press. Soluble Fur proteins were included in the clarified bacterial supernatant. The resulting SsoR inclusion body pellets were solubilized with 20 mM Tris/HCl (pH 7.0), 8 M urea and 200 mM NaCl. SsoR or Fur proteins were further purified by using nickel-ion affinity column (GE Healthcare, Chicago, IL, USA) under denaturing or non-denaturing conditions according to manufacturer instructions. The eluted fractions containing Fur proteins were collected and then concentrated by ultrafiltration (10-kDa cutoff) and exchanged into 20 mM Tris-HCl (pH 8.0) containing 150 mM NaCl. To renature the SsoR protein, the eluted fractions containing SsoR were diluted into 2 M urea, 20 mM Tris/HCl (pH 7.0), 1 mM EDTA by sequential dilutions and then dialyzed against 20 mM Tris/HCl (pH 7.0) overnight. Purified SsoR and Fur proteins were determined by SDS-PAGE and Coomassie brilliant blue staining.

EMSA was performed with the instructions provided in the LightShift Chemiluminescent EMSA Kit (Thermo Fisher Scientific, Rockford, USA). Binding reactions were performed with 40 nmol biotin end-labeled probes, and various amounts of protein in 12 μl binding buffer containing 100 mM Tris/HCl (pH 7.4), 20 mM KCl, 10 mM $MgCl_2$, 2 mM DTT, 40-nmol poly(dI·dC) and 10% glycerol at 15 °C for 60 min. Samples were loaded onto a 6% non-denaturing polyacrylamide gel and subjected to electrophoresis at 80 V for 2 h and then transferred to a nylon membrane (Amersham, Thermo Fisher Scientific, Rockford, USA) in 0.5× TBE at 130 V for 60 min. After UV cross-linking, the probe-protein complexes on the membrane were detected using the Chemiluminescence Nucleic Acid Detection Module Kit (Thermo Fisher Scientific, Rockford, USA).

### Phylogenetic tree construction

A number of UniRef50 representative proteins, including *E. coli* PhoB (*Ec*PhoB), SsoR, *Ec*KdpE, VbrR from *V. parahaemolyticus* (PDB: 7E90), and HP1043 of *H. pylori* (PDB: 2PLN), and their high-sequence-similarity homologs were selected for the analysis. A neighbor-joining phylogenetic tree was constructed using the Clustal W alignment method in MEGA7[88]. The bootstrap consensus tree inferred from 1000 replicates represented the evolutionary history. The EFI Genome-Neighborhood Tool[89] was employed to assay the visualized genomic context of the members in the phylogenetic tree.

## Alignment of the AlphaFold2 predicted and crystal/NMR structures

The AlphaFold2-predicted structures of the proteins included in the phylogenetic tree were obtained from the AlphaFold Protein Structure Database[90]. The number of predicted structures in each cluster in the phylogenetic tree was reduced to no >70 by removing redundant sequences using Jalview[91]. The receiver domains (REC, residue 1–120) of the predicted structures and 69 PDB structures (with each polymer retaining a monomer at random) were aligned in PyMol.

## Structural similarity dendrogram building

Structural similarity dendrogram was built to illustrate the changes in conformations. The method was validated with 55 proteins, which are identical to *Ec*PhoB in sequence, obtained from the AlphaFold Protein Structure Database[90]. The REC domains of these proteins were compared with crystal structures of *Ec*PhoB from the PDB (including 1B00, 1ZES, 2IYN, 2JB9, and 2JBA, with polymers split into monomers). Both AlphaFold2 predicted and PDB monomers exhibited multiple states, with inner, outer, intermediate state A, and intermediate state B as the major states. A structural similarity dendrogram was generated using "All against all" structure comparisons in the DALI server[92], and the resulting dendrogram was visualized using iTOL[93]. Dendrograms for other proteins were generated with AlphaFold2-predicted structures used for structural alignment.

## MD simulations

The dimer complexes of the REC domains of AlphaFold2-predicted SsoR and *Vp*VbrR (PDB: 7E90) were refined by ColabFold[94], and then were applied to MD simulations using CHARMM-GUI[95]. For total eight systems (two mutated dimers and two monomers for SsoR and *Vp*VbrR), a rectangular water box with at least 1 nm edge distance from the protein(s) was used to solve the systems with 150 mM NaCl ions electrolyte to pH 7.0. The periodic boundary conditions were generated for PME FFT by CHARMM-GUI automatically[95]. All-atom CHARMM36m force field was used for ions, protein(s) and TIP3P water and all unbiased simulations were performed using GROMACS-v2023[96]. Before MD production, an energy minimization and the equilibration in the NVT ensemble at a temperature of 310 K using mdp files from CHARMM-GUI were executed sequentially to equilibrate the simulation box. A series of MD simulations were conducted in the NPT ensemble at a temperature of 310 K and a pressure of 1 bar for a total of 3000 ns for each system. Temperature and pressure were coupled using the velocity-rescale method (time constant of 1 ps) and isotropic pressure coupling with the Parrinello-Rahman algorithm (time constant of 5 ps), respectively.

Frames from MD simulation trajectories were processed and extracted using GROMACS, one frame per nanosecond. Structures of different states were grouped based on the relative positions of the switch residue (96 F of SsoR or 98Y of *Vp*VbrR). The positions were described using Chi1 angle of switch residue and the distance between CZ atom of the switch residue and backbone N atom of 99 T (in *Vp*VbrR) or 98 T (in SsoR). The relative positions of the switch residues were analyzed using Plumed-v2.9.0 (developed by PLUMED consortium to promote transparency and reproducibility in enhanced molecular simulations)[97], and RMSF calculations were performed using GROMACS. The 3D structure models and movies were processed and rendered using PyMol.

## Promoter prediction

The multiple promoter prediction tools (BPROM, bTSSfinder, BacPP, and iPromoter-2L) were used to analyze the promoters of the indicated genes[98].

## Statistics and reproducibility

Most analyses were based on a minimum of four independent experiments, yielding biological replicates. Data were shown for either all replicates or presented as mean ± standard error of the mean (SEM). Pairwise comparisons were conducted using Student's *t* test, with a *P* value below 0.05 considered statistically significant. Graphics and statistical analysis were performed using the Prism v9.5.1 software (GraphPad Software LLC, San Diego, CA, USA), completing the statistical test indicated in the text and figure legends.

## Reporting summary

Further information on research design is available in the Nature Portfolio Reporting Summary linked to this article.

## Data availability

The data supporting the findings of this study are available in the article and its supplementary information files. All of the uncropped images in western blotting were shown in Supplementary Fig. 8. The source data underlying the graphs in the paper can be found in Supplementary Data. Supplementary Movie shows the switch residue as it transitions. The raw data for MD are available at https://doi.org/10.5281/zenodo.10924978.

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

## Acknowledgements

This research was supported by the National Natural Science Foundation of China (31930003), the National Key R&D Program of China (2018YFA0901300), and the Natural Science Foundation of Zhejiang Province (LZ23C010002).

## Author contributions

Peilu Xie and Yuanyou Xu, investigation, validation, writing—original draft, formal analysis; Jiaxin Tang, investigation; Shihua Wu and Haichun Gao, writing—review and editing, conceptualization, funding acquisition, project administration, and supervision.

## Competing interests

Haichun Gao is an Editorial Board Member for *Communications Biology*, but was not involved in the editorial review of, nor the decision to publish this article.
