## [Peer Review File · Communications Biology]

Reviewers' comments:

Reviewer #1 (Remarks to the Author):

The manuscript by Xie et al. describes two novel regulations of siderophore biosynthesis system (PubABC) by TCSs in *Shewanella oneidensis*. The first half shows that the BarA/UvrY system regulate the expression of pubABC post-transcriptionally via Csr, and the latter half explains how an orphan RR, SsoR, activates the transcription of pubABC in a phosphorylation-independent manner. Emphasis is placed on bioinformatic analyses using PDB structures and AlphaFold2 structures in order to elucidate how unphosphorylated SsoR can be active. The study is well designed and the results seem to be solid, but the manuscript needs modification. I enjoyed reading the manuscript.

Major comments

1. Line 221 As mentioned here, the expression of each gene was determined from only a single qRT-PCR experiment. Even though three replicates may have been measured, this is only a single data. Thus, statistical analyses between expression shown in Fig. 3B, 3D, 8B and S4B cannot be performed. Results of qRT-PCR experiments tend to vary. At least three biological experiments should be performed for statistical analyses.
2. Lines 585 - 592 The RRs mentioned in this section do not match what is written in the legends of Fig. 7 (lines 1142, 1143). Is Fig. 7D a VbrRM-RD-D51E or VbrRD-RD-D51N? Is Fig. 7F a VbrRM-RD-D51N or VbrRM-RD-D51E? These discrepancies make this section extremely difficult to understand. Please check the manuscript carefully and revise. Also change D51 to D52 for SsoR.
3. When you compare the signal intensity ratio of P-SsoR and NP-SsoR of Fig. 2B and Fig. 4A, SsoR seems to be more phosphorylated in Fig. 4A. How can this happen? Can P-SsoR be dephosphorylated by some HK? Or is this because P-SsoR is not dephosphorylated by HKs? Are there any difference in sample preparation that may cause this difference?
4. Please add results showing loading control (for instance, expression of a house-keeping gene or Ponceau S/CBB stain) for Western blotting results.

Minor comments

1. Line 166 Please explain (spell out) "DFO" here, and not in line 349.
2. What was used for "phosphatase inhibitors"? Please provide the names of the inhibitors.
3. Line 185 Change "mm" to "mM" for 50 mm, 384 mm and 1 mm.
4. Line 232 Was *E. coli* BL21(DE3) used for protein expression? If yes, change BL21 to BL21(DE3) in Table S1 and Fig. S7.
5. Lines 238 - 240 This denaturing step is not written in the legend of Fig. S7. Were SsoR and Fur denatured prior to affinity chromatography?
6. Line 380 The cells were grown to "late-exponential phase" here and "mid-log phase cells" in the legend of Fig. 2 (line 1066). Which is correct?
7. Lines 429, 430 Figure 1E is 1D.
8. Line 441 His-tag to His-tag
9. Line 596 Where is Fig. 5F and alignment results of VxrBs?
10. Line 680 bacterial to bacteria
11. Line 687 strikingly to striking
12. Line 700 VxrBs are not explained.
13. Lines 987 - 996 References 82, 83 and 84 do not appear in the manuscript.
14. Line 1153 Please provide protein concentrations for Fur and SsoR.
15. Figure 2 Are psoR and pso2426 same plasmids? If yes, use the same name.
16. Figure 3 Is the IPTG unit in Fig 3C mM?
17. Figure 6 Is the IPTG unit in Fig 6B and C mM?

Reviewer #2 (Remarks to the Author):

In this manuscript, the authors demonstrate that siderophore production in *Shewanella oneidensis* is governed by the BarA/UvrY-Csr regulatory circuit, the orphan response regulator SsoR, and the iron-responsive transcriptional regulator Fur. They demonstrate that the BarA/UvrY TCS regulates the expression of the siderophore production operon through the small RNAs CsrB1 and CsrB2, along with the translational regulator CsrA, directly influencing pubA translation. Additionally, they shed light on the regulatory behavior of the response regulator SsoR, in which canonical phosphorylation is not required (but eventually takes place) to directly control the transcription of the pub operon. Finally, the authors show that the Fur transcriptional regulator binds to the operator DNA region of both the pub operon and ssoR, effectively repressing their expression in response to iron. These results are significant as they emphasize the versatility of bacterial cells in evolving intricate interactions among environmentally responsive systems, enabling them to better adapt to specific niches. Overall, the manuscript is well-written and the information is presented clearly. The data provided are robust and support the main conclusions.

Major comments

Discussion:

The results from both in vivo gene expression analysis and in vitro EMSA assay provide substantial support for the conclusion that the transcriptional regulation of the pub operon by SsoR does not rely on phosphorylation. Nevertheless, SsoR exhibits phosphorylation capabilities, as evidenced by its detection in a phosphorylated form through Phos-tag SDS-PAGE. As discussed by the authors, this contrasts with *H. pylori* HP1043 RR, which functions in a phosphorylation-independent manner and lacks the phosphorylatable Asp residue. However, the sentence located on page 27, lines 733-736, appears somewhat unclear, prompting the need for the authors to elaborate on the significance of this difference. It would be valuable for the authors to extend their discussion, exploring questions such as whether SsoR-P plays a role in the regulation of other genes, why the apparently unnecessary Asp residue is evolutionarily conserved in SsoR across *Shewanella* species, and whether there are SsoR orthologs that lack the phosphorylation pocket. Addressing these aspects would contribute to a deeper understanding of the implications surrounding the observed phosphorylation dynamics in SsoR.

Minor comments

Title: I find the use of the term "intertwined" in the title somewhat inappropriate, as it may suggest interdependence, whereas the BarA/UvrY and SsoR regulatory effects on siderophore production are, in fact, independent.

Line 106: I find the description of the regulatory processes as "unprecedentedly complex" a bit excessive.

Line 672-674: "Multiple lines of evidence were presented to show that CsrA directly interacts with the pub transcript to block translation." While the results of various genetic experiments presented here strongly suggest direct interaction between CsrA and the pubA transcript to inhibit translation, this interaction has not been directly observed. Therefore, I recommend moderating this statement.

Figure 4: The legend lacks an explanation for the significance of the colored triangles and squares depicted.

Reviewer #3 (Remarks to the Author):

Production of siderophores to acquire sufficient iron is critical for many bacteria. This also applies to species such as *Shewanella* sp., which require quite large amounts of iron for their intricate electron transport chains. Among others, the group of the author previously identified and characterized an intriguing siderophore system in *S. oneidensis*. In this study they now further explored the regulation of siderophore synthesis in this bac. By a mutagenesis approach, they identified two regulators not previously known to be involved in siderophore synthesis, BarA/UvrY/Csr and SsoR. Then using classic genetic approaches they convincingly show the regulation of siderophore synthesis by the first-mentioned system. Intriguingly, SsoR appears to be an orphan response regulator, which occurs in a phosphorylated form in the cell. The authors used an impressive *in silico* analysis to determine the characteristics of SsoR, backed up by genetic/biochemistry approaches to show that this RR belongs to a group of RRs that function without phosphorylation. Furthermore, they showed that, not unexpectedly, the global regulator *fur* is involved in regulation of siderophore production. Based on the results the authors put forward a model of how siderophore production is regulated in *S. oneidensis*.

The authors conducted an impressive amount of work. The manuscript is very clearly structured and it is easy to follow the line of experiments and conclusions. There are a number of intriguing results and I am generally convinced by the experimental evidence.

There is one issue I would like the authors to elaborate on: Why is SsoR still phosphorylated and is there really no difference in the regulation? See 566/6D, while the EMSAs nicely show that the different versions of SsoR readily bind to DNA, the assay (as shown) already indicates differences in affinity. A better assay (maybe SPR) could be used to quantify this. Along the same lines: Figure 6C, while I agree that the mU development upon induced expression looks similar, the D52E variant appears to reach maximum activity earlier, which would fit to the results of the gel shifts. An experiment with more resolution in terms of protein levels would show this. This may indicate that phosphorylation of SsoR may still affect the regulation of this RR (even if subtly). I would also be rather careful with the statement that there is no cognate HK in *S. oneidensis*, I suggest to phrase this a little more carefully where appropriate.

429, if I am not mistaken, CsrA recognizes a small stemloop structure often overlapping with the RBS rather than just small motifs, there would be quite many of those especially in front of start codons. Maybe using one of the many prediction tools out there, the authors can show the mRNA region's secondary structure.

What about the genetic context of *ssrR*? I was intrigued so I looked that up – in *S. oneidensis* there seems to be a putative TonB-like receptor encoded upstream, is this by any chance related to iron uptake? What about the distribution in other species, such as in other *Shewanella* species. How well is the structure conserved?

As the authors state (line 69), siderophore synthesis and transport has been previously shown to be regulated by the GacS/GacA system in *P. aeruginosa*, which is orthologous to *Shewanella* BarA/UvrY. Please elaborate on how the regulation differs (or doesn't) between the two systems and species.

Minor issues, typos:

77, type?

Figure Caption 1D – please indicate the reference for the determination of the Fur binding site.

370, reference(s) describing the differences between the two forms would be good here.

378, please rephrase (‘C-terminally His-tagged’ or so)

430, the RBS

436, add ‘gene’ (or omit ‘the’)

438, please give a range of values here.
441, maybe rather ,higher` or ,more pronounced`
441, His-tag
628, the LacZ reporter

Manuscript ID: COMMSBIO-24-0315 and Multifaceted and intertwined regulation of siderophore synthesis by multiple regulatory systems:

Response to reviewers' comments

Reviewer #1 (Remarks to the Author):

The manuscript by Xie et al. describes two novel regulations of siderophore biosynthesis system (PubABC) by TCSs in *Shewanella oneidensis*. The first half shows that the BarA/UvrY system regulate the expression of pubABC post-transcriptionally via Csr, and the latter half explains how an orphan RR, SsoR, activates the transcription of pubABC in a phosphorylation-independent manner. Emphasis is placed on bioinformatic analyses using PDB structures and AlphaFold2 structures in order to elucidate how unphosphorylated SsoR can be active. The study is well designed and the results seem to be solid, but the manuscript needs modification. I enjoyed reading the manuscript.

We sincerely appreciate your recognition and valuable feedback on our work. We are grateful for the time and effort that you've invested.

Major comments

1. Line 221 As mentioned here, the expression of each gene was determined from only a single qRT-PCR experiment. Even though three replicates may have been measured, this is only a single data. Thus, statistical analyses between expression shown in Fig. 3B, 3D, 8B and S4B cannot be performed. Results of qRT-PCR experiments tend to vary. At least three biological experiments should be performed for statistical analyses.

Response: We have carried out more rounds of qRT-PCR analysis and the results from four biological replicates were presented. Despite the variations among each measurement, statistical analyses confirmed that the differences in expression levels between strains under comparison presented before are confident.

2. Lines 585 - 592 The RRs mentioned in this section do not match what is written in the legends of Fig. 7 (lines 1142, 1143). Is Fig. 7D a VbrRM-RD-D51E or VbrRD-RD-D51N? Is Fig. 7F a VbrRM-RD-D51N or VbrRM-RD-D51E? These discrepancies make this section extremely difficult to understand. Please check the manuscript carefully and revise it. Also, change D51 to D52 for SsoR.

Response: We are very sorry for the misunderstanding caused by the mistakes in the legend of Figure 7. Fig. 7D is VbrR^{M-RD-D51E}, Fig. 7F is VbrR^{D-RD-D51N}. The legend of Figure 7 has been corrected. We also scrutinized the manuscript for mislabeling these proteins and made corrections.

3. When you compare the signal intensity ratio of P-SsoR and NP-SsoR of Fig. 2B and

Fig. 4A, SsoR seems to be more phosphorylated in Fig. 4A. How can this happen? Can P-SsoR be dephosphorylated by some HK? Or is this because P-SsoR is not dephosphorylated by HKs? Are there any difference in sample preparation that may cause this difference?

Response: Clearly, SsoR is present in the cell in both phosphorylated and non-phosphorylated forms, indicating that it can be phosphorylated by kinases or other phosphorylating agents, such as acetyl-phosphate. However, there are two possibilities for coexistence of both phosphorylated and non-phosphorylated SsoR. Phosphorylation may not be effective so that this process could not be complete, or phosphatases carry out dephosphorylation. We also envision a possibility of crosstalk of TCSs. As this study focuses on unravelling the mechanisms underlying phosphorylation-independent regulation of SsoR, we did not exhaustively screen for all possible candidates that phosphorylate or dephosphorylate SsoR.

We assume that the difference in the signal intensity ratio of P-SsoR and NP-SsoR is likely due to variations in sample preparation, especially subtle difference in growth. In addition, despite the same procedure used throughout the study, there are uncontrollable variations in Phos-tag SDS-PAGE and Western blotting experiments. Shown below are the repeating results of P-SsoR and NP-SsoR in WT strain. Based on our data, at this moment, it is certain that SsoR is present in the cell in both phosphorylated and non-phosphorylated forms, but further work is needed to determine the ratio. We are currently screening for agents affecting phosphorylating state of SsoR. After we identify them we believe that we are able to evaluate the difference in the signal intensity ratio of P-SsoR and NP-SsoR.

4. Please add results showing loading control (for instance, expression of a house-keeping gene or Ponceau S/CBB stain) for Western blotting results.

Response: We provided CBB stains for some of Western blotting results in the supplementary information for review. We always load the same amount of proteins for each lane in one experiment.

Minor comments

1. Line 166 Please explain (spell out) "DFO" here, and not in line 349.

Response: "DFO" has been spelled out at Line 166.

2. What was used for "phosphatase inhibitors"? Please provide the names of the inhibitors.

Response: We are sorry that we can't provide the names of the inhibitors because the company (Solarbio, Beijing, China) that manufactures the phosphatase

inhibitors refuses to inform the specific components. We are informed that the inhibitors are used in a mixture of protease and phosphatase inhibitors for bacterial protein extraction. It contains a broad spectrum of serine, cysteine, and acid protease inhibitors/aminopeptidase inhibitors, and serine/threonine, tyrosine, acid, and alkaline phosphatase inhibitors. As we know, Merck provides information on the ingredients of their phosphatase Inhibitors, which can be used as a reference (<https://www.sigmaaldrich.cn/CN/en/products/protein-biology/protein-sample-prep/inhibitor-cocktails#phosphataseprotease>).

3. Line 185 Change “mm” to “mM” for 50 mm, 384 mm and 1 mm.

Response: Thank you for your attentiveness. These units have been corrected.

4. Line 232 Was *E. coli* BL21(DE3) used for protein expression? If yes, change BL21 to BL21(DE3) in Table S1 and Fig. S7.

Response: Yes, the strains used for the purifying proteins in this study were *E. coli* BL21 (DE3). We made changes accordingly.

5. Lines 238 – 240 This denaturing step is not written in the legend of Fig. S7. Were SsoR and Fur denatured prior to affinity chromatography?

Response: The denaturing step has been added in the legend of Fig. S7. As described in the Materials and Methods section, the purification conditions for Fur protein were natural. In contrast, SsoR proteins were in inclusion bodies at *E. coli* BL21 (DE3), and the buffer used for affinity chromatography was supplemented with the denaturant 8M urea. SsoR proteins were concentrated and denatured after affinity chromatography.

6. Line 380 The cells were grown to “late-exponential phase” here and “mid-log phase cells” in the legend of Fig. 2 (line 1066). Which is correct?

Response: “mid-log phase cells” is correct. More precisely, cells grown to an OD₆₀₀ of 0.6 were collected.

7. Lines 429, 430 Figure 1E is 1D.

8. Line 441 His-tag to His-tag

Response: We made corrections.

9. Line 596 Where is Fig. 5F and alignment results of VxrBs?

Response: We are very sorry for miswriting 5D as 5F. The second subplot of the Figure. 5D shows the alignment results for VxrBs. It has been corrected.

10. Line 680 bacterial to bacteria

11. Line 687 strikingly to striking

Response: These have been revised accordingly.

12. Line 700 VxrBs are not explained.

Response: We have used VbrRs to replace VxrBs as the representative of the cluster. By doing this, all confusions VxrBs brought up are eliminated.

13. Lines 987 – 996 References 82, 83 and 84 do not appear in the manuscript.

Response: These references have been removed. The reference numbers have been updated throughout the article.

14. Line 1153 Please provide protein concentrations for Fur and SsoR.

Response: The protein concentrations used for the EMSA assays have been described in the legend of Figures 8A and 8E.

15. Figure 2 Are pssor and pso2426 same plasmids? If yes, use the same name.

Response: Yes, pssor and pso2426 are the same plasmids. Figure 2 has been revised.

16. Figure 3 Is the IPTG unit in Fig 3C mM?

17. Figure 6 Is the IPTG unit in Fig 6B and C mM?

Response: Yes, the IPTG unit is mM. Figure 3C and Figure 6 have been corrected.

Reviewer #2 (Remarks to the Author):

In this manuscript, the authors demonstrate that siderophore production in *Shewanella oneidensis* is governed by the BarA/UvrY-Csr regulatory circuit, the orphan response regulator SsoR, and the iron-responsive transcriptional regulator Fur. They demonstrate that the BarA/UvrY TCS regulates the expression of the siderophore production operon through the small RNAs CsrB1 and CsrB2, along with the translational regulator CsrA, directly influencing pubA translation. Additionally, they shed light on the regulatory behavior of the response regulator SsoR, in which canonical phosphorylation is not required (but eventually takes place) to directly control the transcription of the pub operon. Finally, the authors show that the Fur transcriptional regulator binds to the operator DNA region of both the pub operon and ssoR, effectively repressing their expression in response to iron. These results are significant as they emphasize the versatility of bacterial cells in evolving intricate interactions among environmentally responsive systems, enabling them to better adapt to specific niches.

Overall, the manuscript is well-written and the information is presented clearly. The data provided are robust and support the main conclusions.

Thank you for your positive evaluation and constructive comments.

Major comments

Discussion:

The results from both in vivo gene expression analysis and in vitro EMSA assay provide substantial support for the conclusion that the transcriptional regulation of the *pub* operon by SsoR does not rely on phosphorylation. Nevertheless, SsoR exhibits phosphorylation capabilities, as evidenced by its detection in a phosphorylated form through Phos-tag SDS-PAGE. As discussed by the authors, this contrasts with *H. pylori* HP1043 RR, which functions in a phosphorylation-independent manner and lacks the phosphorylatable Asp residue. However, the sentence located on page 27, lines 733-736, appears somewhat unclear, prompting the need for the authors to elaborate on the significance of this difference. It would be valuable for the authors to extend their discussion, exploring questions such as whether SsoR-P plays a role in the regulation of other genes and why the apparently unnecessary Asp residue is evolutionarily conserved in SsoR across *Shewanella* species, and whether there are SsoR orthologs that lack the phosphorylation pocket. Addressing these aspects would contribute to a deeper understanding of the implications surrounding the observed phosphorylation dynamics in SsoR.

Response: Thanks for this excellent comment. The concerns enclosed are what we endeavor to address now, which will be extended to all types of TCS RRs. The work will be a combination of structures, computational modelling, and functional characterization. Currently, determination of the structures of SsoR and relevant mutants is still ongoing. As this manuscript is heavily packed, we would like to present the results in a separate communication soon. Following is the closely related content to the comment.

SsoR retains the ability to be phosphorylated but does not rely on phosphorylation to regulate transcription of target genes. We demonstrated that the Fur responds to intracellular iron concentrations, acting as an upstream regulator of *ssrR*. This leads us to propose that the unusual feature (phosphorylation-independent activity with phosphorylation residue) of SsoR may be a consequence of either Fur regulation or the accidental loss of a HK. Hence, we think the preservation of the seemingly redundant phosphorylation site in SsoR, unlike in HP1043, is largely due to evolutionary diversity. To explore whether SsoR regulates other iron homeostasis-related genes beyond *pub* and *ssrR* in *S. oneidensis*, we are conducting proteomic and transcriptomic analyses. The differentially expressed genes and proteins under iron-limited conditions between WT and *ssrR* deleted mutant strains were identified. Some of them are currently being examined with EMSA. From the preliminary data, we are certain that SsoR directly controls transcription of genes other than those we included in this study.

The phosphorylation state of the phosphorylatable Asp residue notably influences the dimerization of RDs in typical RR (7E90). This effect was elucidated through our calculations of residue interaction frequencies and the binding energy between monomers. Our observations indicate that the interaction frequencies of residues located at the dimerization interface were reduced in 7E90^{D51N} (Fig. 1), resulting in a decreased binding energy (using *gmx_MMGBSA*) with a value of -53.43 ± 11.82 kcal/mol for 7E90^{D51N} as compared to -62.90 ± 10.32 kcal/mol for 7E90^{D51E} (Fig. 2).

Additionally, we conducted 3ns MD simulations for 7E90^{D51Q}. While glutamine's side chain only contains one more carbon atom than asparagine's, the $\alpha 4$ helix underwent a conformational change, resulting in its transformation from a helix to a loop. And we thought 7E90^{D51Q} is unable to perform normal physiological functions.

Fig 1. The contact map for 7E90.

For the Fig 1:

At first, we define a contact to be formed if: (1) the distance between the closest atoms of residues is shorter than 4 Å, (2) the residues are positioned at the dimerization interface (residues 83-166), with each residue in a pair originating from a different monomer, and (3) the contact is present for more than 30% of the simulation time, ensuring that we focus on positive contacts.

Secondly, the contact pairs for 7E90^{D51E} were visualized in heatmap (Fig 1. A), where the x-axis and y-axis represent residues from different monomers, with the corresponding protein residue locations indicated on the left. The color gradient ranging from white to yellow and red reflects the contact frequency (e.g., a value of 100 indicates that the contact was present 100% of the time during the simulation). Additionally, we calculated the cumulative frequency for each residue in 7E90^{D51E} and 7E90^{D51N}, and represented it in the form of a histogram attached to the heatmap. Cumulative frequency for 7E90^{D51E} are shown in blue, while residues for 7E90^{D51N} are shown in orange.

Then, to illustrate the contact frequency change in each contact pair, we conducted Fig 1. B. The map is similar to Fig 1. A, but the colors represent the increase (in red) or decrease (in blue) of contact frequency in 7E90^{D51N} compared to 7E90^{D51E}.

Fig. 2 The binding free energy of typical and atypical RRs.

In SsoR, the interaction frequencies of residues located at the dimerization interface did not exhibit a pronounced reduction or increase between SsoR^{D52E} and SsoR^{D52N} (Fig. 3). And this observation was further supported by the results of gm_x_MMGBSA.(Fig. 2).

Fig. 3 The contact map for SsoR. Same as Fig. 1, A depicts the contact pairs for SsoR^{D51E}, and B represents the increase (in red) or decrease (in blue) of contact frequency in SsoR^{D51N} compared to SsoR^{D51E}.

Minor comments

Title: I find the use of the term "intertwined" in the title somewhat inappropriate, as it may suggest interdependence, whereas the BarA/UvrY and SsoR regulatory effects on siderophore production are, in fact, independent.

Response: We removed it.

Line 106: I find the description of the regulatory processes as "unprecedentedly complex" a bit excessive.

Response: We agree with the comment and remove the "unprecedentedly".

Line 672-674: "Multiple lines of evidence were presented to show that CsrA directly interacts with the pub transcript to block translation." While the results of various genetic experiments presented here strongly suggest direct interaction between CsrA and the pubA transcript to inhibit translation, this interaction has not been directly observed. Therefore, I recommend moderating this statement.

Response: The sentence has been changed to "Multiple lines of evidence were presented to support that CsrA directly interacts with the pub transcript to block translation."

Figure 4: The legend lacks an explanation for the significance of the colored triangles and squares depicted.

Response: We have added required information.

Reviewer #3 (Remarks to the Author):

Production of siderophores to acquire sufficient iron is critical for many bacteria. This also applies to species such as *Shewanella* sp., which require quite large amounts of iron for their intricate electron transport chains. Among others, the group of the author previously identified and characterized an intriguing siderophore system in *S. oneidensis*. In this study they now further explored the regulation of siderophore synthesis in this bac. By a mutagenesis approach, they identified two regulators not previously known to be involved in siderophore synthesis, BarA/UvrY/Csr and SsoR. Then using classic genetic approaches they convincingly show the regulation of siderophore synthesis by the first-mentioned system. Intriguingly, SsoR appears to be an orphan response regulator, which occurs in a phosphorylated form in the cell. The authors used an impressive in silico analysis to determine the characteristics of SsoR, backed up by genetic/biochemistry approaches to show that this RR belongs to a group of RRs that function without phosphorylation. Furthermore, they showed that, not unexpectedly, the global regulator fur is involved in regulation of siderophore production. Based on the results the authors put forward a model of how siderophore production is regulated in *S. oneidensis*.

The authors conducted an impressive amount of work. The manuscript is very clearly structured and it is easy to follow the line of experiments and conclusions. There are a number of intriguing results and I am generally convinced by the experimental evidence.

Thank you for acknowledging our research and providing insightful feedback. We will carefully consider each recommendation to enhance our manuscript. Your expertise and time are greatly appreciated.

There is one issue I would like the authors to elaborate on: Why is SsoR still phosphorylated and is there really no difference in the regulation? See 566/6D, while the EMSAs nicely show that the different versions of SsoR readily bind to DNA, the assay (as shown) already indicates differences in affinity. A better assay (maybe SPR) could be used to quantify this. Along the same lines: Figure 6C, while I agree that the mU development upon induced expression looks similar, the D52E variant appears to reach maximum activity earlier, which would fit to the results of the gel shifts. An experiment with more resolution in terms of protein levels would show this. This may indicate that phosphorylation of SsoR may still affect the regulation of this RR (even if subtly). I would also be rather careful with the statement that there is no cognate HK in *S. oneidensis*, I suggest to phrase this a little more carefully where appropriate.

Response: Thanks for this excellent comment. The regulatory ability of SsoRs in different phosphorylation states may indeed vary, a notion that is also supported by molecular dynamics simulations. In SsoR^{M-RD-D52N} (Fig. 7G), a small amount of outer state appears, which is not observed in SsoR^{M-RD-D52E} (Fig. 7C). In addition, the RMSF of the switch residue in SsoR^{M-RD-D52N} is more flexible. These suggest that SsoR^{M-RD-D52E} may be more inclined towards dimer formation than SsoR^{M-RD-D52N} (We modified the content of lines 622-626). As for the specific mechanisms involved, investigations are currently ongoing and results will be presented in a separate communication (Please refer to our response to the major comment of Reviewer #2). We believe that we would be able to address the concerns mentioned in this comment fully with new data.

The orphan kinases are the most likely candidates, but our experiments have confirmed that they are not cognate HK to SsoR, at least not exclusive cognate HK. However, we do not rule out the possibility of interactions between SsoR and multiple alternative orphan kinases or non- cognate kinases. Therefore, we agree with the reviewer such that we have adopted a more cautious phrasing (Line 718-724).

429, if I am not mistaken, CsrA recognizes a small stemloop structure often overlapping with the RBS rather than just small motifs, there would be quite many of those especially in front of start codons. Maybe using one of the many prediction tools out there, the authors can show the mRNA region's secondary structure.

Response: The consensus sequence for CsrA binding site was determined as RUACARGGAUGU, with a conserved GGA motif that is typically located in a hexaloop (ARGGAU) of an RNA hairpin(1). Indeed, in addition to the AAGGAG that overlaps with the SD sequence, other predicted CsrA binding sites were found in the untranslated and coding regions of *pub* mRNA. In this study, we only verified

the importance of AAGGAG. Tony Romeo et al. reviewed various regulatory mechanisms of CsrA (2). Our study validates the most common mechanism.

During our investigation, we predicted the secondary structure of *pub* mRNA but did not display the result in the manuscript for multiple reasons. First, as shown in the figure below, several RNA fold structure prediction tools, including Vienna RNA(3) and Mfold(4), yielded significantly different secondary structure predictions. Lacking expertise in RNA structural biology, we found it challenging to assess the accuracy of these predictions and to correctly interpret the information of the secondary structure *pub* mRNA. Second, our mutational analysis indicated the binding of CsrA to the 5' untranslated region and that the interaction of CsrA with the AAGGAG sequence plays a key role. However, we did not perform RNA EMSA experiments or CsrA-mRNA footprints to identify the exact binding sites of CsrA. Third but not the least, this paper focuses on SsoR, and our findings with CsrA do not surprise us, we just move on. Please advise us if there is something new about the regulation. If so, we would love to discuss this or work with you on this.

What about the genetic context of *ssoR*? I was intrigued so I looked that up – in *S. oneidensis* there seems to be a putative TonB-like receptor encoded upstream, is this by any chance related to iron uptake? What about the distribution in other species, such as in other *Shewanella* species. How well is the structure conserved?

Response: SO_2427 is a TonB-dependent receptor composed of an N-terminal plug domain and a C-terminal β -barrel domain, similar to vitamin B12 transporters (BtuB) and siderophore transporter (FhuE, FepA). Although the specific substrate of SO_2427 remains unknown, it may not play a role in iron uptake because PutA is the one important for siderophore-mediated iron uptake. The genetic context of *ssoR* in *Shewanella* species and other bacteria is shown below. We retrieved *ssoR* homologs from the *Pseudomonas* phylum and constructed a phylogenetic tree, identifying five distinct branches for *Shewanella ssoR* homologs. The genomic context revealed that cluster I homologs are associated with a TonB-dependent receptor and a Carboxypeptidase, similar to *ssoR*. Cluster II homologs are orphan RRs with a distinct genomic context from *ssoR*. More distantly related clusters (III, IV, V) exhibit different genomic contexts and are adjacent to HK genes. SO_2427 is conserved in cluster I but not conserved in other clusters. Additionally, these data

suggest that the ancestor of SsoR had paired HKs, with some descendants losing their HKs over evolution to become orphan RRs. Due to the limited length of the article, we did not include the evolutionary of the *ssorR* homologs of *Shewanella* species.

As the authors state (line 69), siderophore synthesis and transport has been previously shown to be regulated by the GacS/GacA system in *P. aeruginosa*, which is orthologous to *Shewanella* BarA/UvrY. Please elaborate on how the regulation differs (or doesn't) between the two systems and species.

Response: GacS/GacA system in *P. aeruginosa* is orthologous of *Shewanella* BarA/UvrY system. Our research indicates that the mechanism by which *Shewanella*'s BarA/UvrY system regulates siderophore synthesis is similar to that in *E. coli* and *P. aeruginosa*, utilizing the traditional positive regulatory cascade of BarA-UvrY/Csr-CsrA (Gac/Rsm-RsmE). It has been found that the regulatory mechanism and pathway of the Gac/Rsm cascade in *Pseudomonas protegens* H78 are complex, with two homologs of CsrA (RsmA and RsmE) and three corresponding sRNAs (RsmXYZ). Besides the Gac/Rsm-RsmE traditional positive regulatory cascade, *P. protegens* H78 has RsmA-mediated positive transcriptional regulation(5). Following your suggestion, we have added a statement in the discussion section (Line 674-676), without elaborating on the regulatory characteristics of *P. protegens*.

Minor issues, typos:

77, type?

Response: We rephrased the sentence.

Figure Caption 1D – please indicate the reference for the determination of the Fur binding site.

Response: In our previous article titled ‘Dissociation between Iron and Heme Biosyntheses Is Largely Accountable for Respiration Defects of *Shewanella oneidensis fur* Mutants’, the Fur box and the target gene of Fur were predicted. We cite this article and refer to Figure 1D in the main text section (Line 614-617).

370, reference(s) describing the differences between the two forms would be good here.

Response: We have added two references describing winged helix-turn-helix DNA-binding domain and helix-turn-helix DNA-binding domain, respectively (Line 370-371).

378, please rephrase (,C-terminally His-tagged’ or so)

430, the RBS

436, add ,gene’ (or omit ,the’)

Response: We made changes accordingly.

438, please give a range of values here.

Response: We calculated the ratio of *pubA*'s mRNA levels in these strains and the WT strain to be 0.78 ($\Delta barA$), 0.77 ($\Delta uvrY$), 0.81 ($\Delta csrB1\Delta csrB2$), 1.15 ($\Delta csrA$), 1.20 ($\Delta barA\Delta csrA$), 1.93 ($\Delta uvrY\Delta csrA$), respectively. Based on the calculations, we made modifications (Line 439-440).

441, maybe rather ,higher’ or ,more pronounced’

Response: ‘more significant’ has been revised to ‘more pronounced’.

441, His-tag

Response: It has been revised.

628, the LacZ reporter

Response: Throughout the manuscript, the ‘*lacZ* reporter’ has been revised to ‘LacZ reporter’.

Rererences:

1. Dubey, A.K., Baker, C.S., Romeo, T. and Babitzke, P. (2005) RNA sequence and secondary structure participate in high-affinity CsrA-RNA interaction. *RNA*, **11**, 1579-1587.
2. Romeo, T. and Babitzke, P. (2018) Global Regulation by CsrA and Its RNA Antagonists. *Microbiol Spectr*, **6**, RWR-0009-2017.
3. Gruber, A.R., Lorenz, R., Bernhart, S.H., Neuböck, R. and Hofacker, I.L. (2008) The Vienna RNA websuite. *Nucleic Acids Res.*, **36**, W70-W74.
4. Zuker, M. (2003) Mfold web server for nucleic acid folding and hybridization prediction. *Nucleic Acids Res.*, **31**, 3406-3415.

5. Wang, Z., Huang, X., Liu, Y., Yang, G., Liu, Y. and Zhang, X. (2017) GacS/GacA activates pyoluteorin biosynthesis through Gac/Rsm-RsmE cascade and RsmA/RsmE-driven feedback loop in *Pseudomonas protegens* H78. *Mol. Microbiol.*, **105**, 968-985.

REVIEWERS' COMMENTS:

Reviewer #1 (Remarks to the Author):

My concerns have been appropriately responded, and I look forward to the future work of this group.

Reviewer #2 (Remarks to the Author):

In this revised manuscript all of my concerns have been effectively addressed, and I believe the new version of the manuscript is ready for acceptance and publication.

Reviewer #3 (Remarks to the Author):

The authors have adequately responded to my issues. I don't have any further comments.